# Perceptual Score: What Data Modalities Does Your Model Perceive?

**Itai Gat**
Technion

**Idan Schwartz**
Technion
NetApp

**Alexander Schwing**
University of Illinois at Urbana-Champaign

## Abstract

Machine learning advances in the last decade have relied significantly on large-scale datasets that continue to grow in size. Increasingly, those datasets also contain different data modalities. However, large multi-modal datasets are hard to annotate, and annotations may contain biases that we are often unaware of. Deep-net-based classifiers, in turn, are prone to exploit those biases and to find shortcuts. To study and quantify this concern, we introduce the perceptual score, a metric that assesses the degree to which a model relies on the different subsets of the input features, *i.e.*, modalities. Using the perceptual score, we find a surprisingly consistent trend across four popular datasets: recent, more accurate state-of-the-art multi-modal models for visual question-answering or visual dialog tend to perceive the visual data less than their predecessors. This trend is concerning as answers are hence increasingly inferred from textual cues only. Using the perceptual score also helps to analyze model biases by decomposing the score into data subset contributions. We hope to spur a discussion on the perceptiveness of multi-modal models and also hope to encourage the community working on multi-modal classifiers to start quantifying perceptiveness via the proposed perceptual score.

## 1 Introduction

Machine learning advances over the last decade are remarkable. Challenges that seemed daunting merely ten years ago are now a breeze, and new applications that we barely dared to dream about seem achievable within the next few years. Indeed, accuracy metrics on tasks like visual question answering and reasoning suggest significant improvements.

Reported improvements are to a large extent due to the availability of large datasets [1–3], computational performance advances, *e.g.*, for GPUs, and a better understanding about how to encode inductive biases into deep-nets, *e.g.*, by using rectified linear units [4], normalization [5], skip connections [6], transformers [7], *etc*. However, importantly, developed deep-net architectures are not guaranteed to solve a given task. There is a chance that they may instead exploit dataset biases.

This concern is surely in part due to non-robust training techniques, and a plethora of methods improve classifier robustness [8–10]. However, datasets play an important role in controlling the extracted bias as well. For instance, if correct answers in a question-answering task are significantly shorter than incorrect ones, classifier training should not use answer length as a cue. Although this seems reasonable, for audio-visual scene aware dialog, Schwartz et al. [11] find for example that in many cases the question alone is sufficient to generate a scene-aware dialog response, avoiding the need to look at the video. Hence, in order to assess the suitability of a classifier, we need to understand how much it relies on different data modalities.

To quantify how much a classifier relies on its different input modalities, we introduce the perceptual score. The perceptual score assesses the degree to which a model relies on a modality. To do so the perceptual score permutes the features of a modality across samples in the test set after the classifier

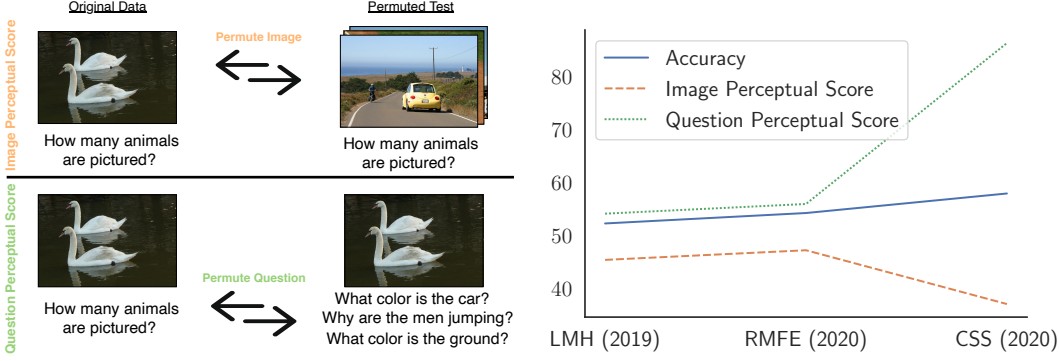

(a) Visual question answering (VQA) data. Modality perceptiveness is measured through a permutation test.

(b) Progress of VQA-CP models.

Figure 1: Multi-modal datasets often have undesired biases: (a) To identify those biases we suggest the perceptual score as a new metric. It assesses the change in prediction when a model's input for some modalities is permuted during testing. If the classifier output remains identical despite permutation, a model doesn't perceive the modality. (b) Using the perceptual score we identify that recent progress of VQA models may not be entirely due to better reasoning.

was trained, as illustrated in Fig. 1a. If the classifier's performance drops to or below chance level, the perceptual score is high. This intuitively applies to single-modality models too: randomly permuting test data and labels after training results in chance-level classification accuracy.

Using the perceptual score, we find a surprisingly consistent trend across five popular datasets (VQA, VQA-CP, VisDial, SocialIQ, DISCO): recent, more accurate state-of-the-art multi-modal models for visual question-answering or visual dialog tend to perceive the visual data less than their predecessors (see Fig. 1b). This trend is concerning as answers are hence increasingly inferred from textual cues only. Using the perceptual score also helps to analyze model biases by decomposing the score into data subset contributions. For example, the perception of an image and question varies depending on the question type. None of the recent VQA-CP models showed high image perception scores for 'number'-type questions. A surprisingly low image perception score is obtained for the state-of-the-art model when confronted with 'yes/no' questions.

We hope the perceptual score spurs a discussion regarding the perceptiveness of multi-modal models and we also hope to encourage the community working on multi-modal classifiers to start quantifying perceptiveness of models.

Our contributions:

- We propose the perceptual score, a simple yet effective method for assessing the perceptiveness of multi-modal models towards a modality.
- Our experiments span multiple datasets and models. We find that multi-modal models tend to ignore some modalities while taking shortcuts.
- Subsequently, we investigate the sources of bias on popular multi-modal datasets such as VQA-CP and SocialIQ: We find that SocialIQ is biased by sentiment, and bias in the VQA-CP model results from shifting the training priors to more closely resemble those in the test set.

## 2 Related Work

The perceptual score assesses the degree to which a classifier relies on a particular input modality. This is related to studying datasets and their biases, methods which aim to reduce the biases captured by classifiers and work which studies the importance of features. We review all three areas next.

**Datasets and bias:** Data has been a central element for machine learning progress [12–14] in the last two to three decades. The ImageNet challenge [2] and the development of AlexNet [15] sparked the deep learning era. But as datasets grow, biases emerge which may go undetected for a long time. For instance, the background in ImageNet can reveal information about the object class [16]. Also, with the increasing popularity of crowdsourcing systems like Amazon Mechanical Turk, many datasets

are annotated in uncontrolled environments. Different annotators are hence injecting unknown socio-economic properties into dataset annotations [17]. Those dataset biases can be detrimental to the considered task. For instance, meticulously collected visual question answering (VQA) data [1] aims to provide a platform for exciting research to advance image-language understanding. However, it is non-trivial to remove biases from this type of data. Indeed, it was reported that the question solely is sufficient to detect the correct answer [18, 19], *i.e.*, no image information is required. In an attempt to fix this bias, the dataset was re-annotated [20], or the train and test split were re-organized [21]. Similarly, Schwartz et al. [11] reported that in Audio-visual-scene-aware dialog (AVSD) [22], the question cue is often stronger, making the desired video and sound reasoning implausible or unnecessary. Likewise, SNLI [23], aims to determine the correctness of a hypothesis given a premise. However, Gururangan et al. [24] point out that an internal bias exists: linguistic features not related to the premise correlate with the label. Further, the Story Cloze [25] dataset permits to develop models which estimate the correct ending of a story. Schwartz et al. [26] show that, for this dataset, length alone is a powerful feature to determine the correct ending. Teney et al. [27] discuss the pitfalls of current out-of-distribution VQA methods. Perceptual score provide a way of identifying and quantifying these limitations.

Recently, Zadeh et al. [28] proposed SocialIQ, a dataset intended to reason about social situations in videos, specifically, emotion detection in a social situation. In this dataset, given a video and a question, the correct social situation should be recognized, *e.g.*, "The man is upset because he is being insulted." We show that it is easy to pick the correct statement using only the text data. This is possible because the statement's correctness often correlates with the statement's sentiment. Again, much like for VQA and AVSD, image information doesn't seem to be necessary for reasonably accurate performance. Hence, biases exist which permit to 'address the dataset' without addressing the task. For SocialIQ, we attribute these biases to the fact that social reasoning is considered difficult, even for humans. Hence an annotator's expertise is particularly important.

Importantly, going forward, we think it is elusive that we will be able to create unbiased datasets. We hence need techniques to automatically measure biases. In this work, we provide a mechanism that permits to do this for any dataset.

**Methods to reduce bias:** Several methods have been suggested to reduce bias from a dataset. Some techniques require prior knowledge of the biased variables, for instance, gender bias in vision is addressed by masking related features (*e.g.*, faces) [29–38]. Also, some techniques require access to the test set to re-balance it [39]. Additionally, various approaches were proposed for visual question answering [40–43]. These methods use a classifier trained only on the question modality to regularize bias directly. Cadene et al. [40] suggest to mask a model's softmax prediction with a softmax prediction of a subset classifier trained on the question modality. Clark et al. [41] use an ensemble method of a full classifier paired with a question subset classifier using products of experts [44]. Common to all those approaches is the use of a subset classifier to indicate reliance on subset of the data. In contrast, we assess the perceptiveness of the original model based on permutations of a subset of the data.

**Feature importance methods:** Also relevant to our work are techniques that measure feature-importance. These works generally focus on understanding why a model made a particular prediction. To this end, Ribeiro et al. [45] introduce LIME, a local explanation method that approximates a linear explanation model around a given example. Later, Shrikumar et al. [46] proposed DeepLift specifically for deep net models. Lundberg and Lee [47] introduced SHapley Additive exPlanations (SHAP), a method to estimate feature importance using an approximation of the features Shapley values. LIME assumes that the local model is linear, while SHAP does not have any such assumptions. Lundberg and Lee [47] show a connection between DeepLIFT and Shapely values introduced as 'Deep SHAP.' Each of those techniques concentrates on the importance of features, whereas we focus on the importance of a modality as a whole. Hessel et al. [48] proposed to assess how much a classifier benefits from interactions across different data modalities. To do this, EMAP approximates a multi-modal classifier with classifiers that each depend on only a single data modality. In contrast, we measure the perceptiveness of each modality separately and for every data point independently, *i.e.*, we aim to identify how much a classifier relies on each data modality.

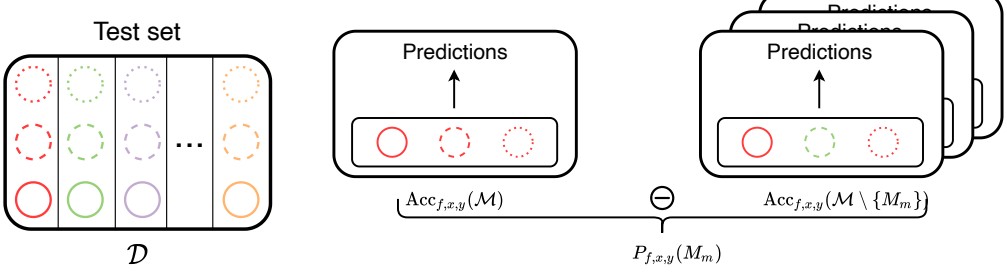

(a) Three modalities data.  (b) Calculation of sample perceptual score.

Figure 2: Calculation of sample perceptual score: (a) Shows a three-modality test set where each sample is colored differently. A different line style distinguishes modalities. (b) Demonstrates how to compute the sample perceptual score. Given a test set $\mathcal{D}$, we measure the sample $x$ accuracy of the original model on the left. We calculate the average accuracy across several permuted samples on the right (see Eq. (3)). The perceptual sample score is the difference of those two terms (see Eq. (2)).

# 3 The Perceptual Score

The *perceptual score* quantifies the degree to which a model relies on the input data or a subset thereof. Said differently, the perceptual score assesses the importance of data or a subset of the data for a model's result. For instance, if a model answers questions about an image without using cues extracted from the image modality, the model does not perceive the image modality. In this case we want the perceptual score for the image modality to be lower than that of a model which relies heavily on the image modality.

We believe that reporting the perceptual score of a model in addition to its accuracy is particularly important in a multi-modal setting. As a community we are developing increasingly complex models. However, to date we know very little about what parts of the data these models rely on. We think this lack in our understanding is due to a missing easy way to quantify how much a model relies on available data modalities. To rectify this we hope to illustrate that a simple yet intuitive score like the proposed perceptual score is very useful and easy to report too.

The following sections describe the perceptual score of a modality for a given model and dataset.

## 3.1 Setup

Let $\mathcal{D} = \{(x_1, y_1), \ldots, (x_{|\mathcal{D}|}, y_{|\mathcal{D}|})\}$ denote the test set data, where $|\mathcal{D}|$ refers to the number of samples in the dataset. Each sample is a pair consisting of the input data $x_i \in \mathcal{X}$ and its corresponding label $y_i \in \mathcal{Y}$. Here, $\mathcal{Y}$ denotes a finite set of possible classes. In multi-modal datasets which we consider here, the data $x_i$ can be separated naturally into different parts as illustrated in Fig. 2a. For instance, in the SocialIQ task, we can partition the data into video-related, question-related and answer-related parts. Formally, let $\mathcal{M} = \{M_1, \ldots, M_{|\mathcal{M}|}\}$ be a set of modalities of size $|\mathcal{M}|$, *e.g.*, the video-, the question- and the answer-modality. We partition the data $x_i$ into its modalities using a set notation, *i.e.*, $x_i = \{x_i^{M_1}, \ldots, x_i^{M_{|\mathcal{M}|}}\}$. We use $x_i^{\{M_1, M_2\}} = \{x_i^{M_1}, x_i^{M_2}\}$ to refer to the first two modalities, *i.e.*, the superscript can be a set.

## 3.2 Perceptual Score of a Data Modality

The perceptual score $P_{f,\mathcal{D}}(M_m)$ of a model $f : \mathcal{X} \to \mathcal{Y}$ towards modality $M_m$ on data $\mathcal{D}$ is defined as

$$P_{f,\mathcal{D}}(M_m) = \frac{1}{Z} \left( \mathbb{E}_{(x,y)\sim\mathcal{D}}[P_{f,x,y}(M_m)] \right), \tag{1}$$

*i.e.*, as the normalized expectation of sample perceptual scores $P_{f,x,y}(M_m)$. Here, $Z$ indicates the normalization factor, which can either be determined by the dataset alone (*i.e.*, $Z = Z_{\mathcal{D}}$) or based on both the dataset and the model (*i.e.*, $Z = Z_{f,\mathcal{D}}$). We discuss the normalization in Sec. 3.2.2.

The sample perceptual score $P_{f,x,y}(M_m)$, outlined in Fig. 2b, aims to measure the degree to which a model $f : \mathcal{X} \to \mathcal{Y}$ relies on a modality $M_m$ for prediction of sample $x \in \mathcal{X}$. To do so we define the

sample perceptual score for modality $M_m \in \mathcal{M}$ as the normalized difference between the accuracy of a model which uses all data modalities, and the accuracy of a model which doesn't use modality $M_m$ for prediction, *i.e.*, as

$$P_{f,x,y}(M_m) = \text{Acc}_{f,x,y}(\mathcal{M}) - \text{Acc}_{f,x,y}(\mathcal{M} \setminus \{M_m\}). \tag{2}$$

Here, $\mathcal{M} \setminus \{M_m\}$ is an operator that removes the influence of modality $M_m$ from the set of all modalities $\mathcal{M}$. We define this operation formally in Sec. 3.2.1. Note, $\text{Acc}_{f,x,y}(\mathcal{M})$ refers to the classical prediction accuracy of a dataset sample $(x, y)$ for a given trained model $f$ with $\mathcal{M}$ the set of all modalities used for prediction, *i.e.*, $\text{Acc}_{f,x,y}(\mathcal{M}) = \mathbb{1}_{f(x^{\mathcal{M}})=y}$.

Intuitively, the sample perceptual score $P_{f,x,y}(M_m)$ is high if the accuracy of a model that does not consider modality $M_m$ for prediction is significantly smaller than the accuracy of a model which uses all data modalities $\mathcal{M}$. Conversely, if the accuracy doesn't change, irrespective of whether modality $M_m$ is available or not, the model $f$ doesn't perceive the modality $M_m$. Note that in cases where the modality $M_m$ irritates the model, the perceptual score can be negative.

In the following we discuss how to 'remove' a modality $M_m$ from a model $f$ (Sec. 3.2.1) and how to compute the normalization constant $Z$ (Sec. 3.2.2).

### 3.2.1 Removing Modality Influence

Removing a modality from a trained model is difficult since typical models entangle modalities and compute high-order correlations. Ideally, we need a tool that minimizes the impact of one modality while maintaining the other components' functionality. To achieve this, we study a permutation-based approach, *i.e.*, we randomly permute the modality-related features among the test set data $\mathcal{D}$. We think permutation is particularly useful for the perceptual score because it is hyper-parameter free. This ensures that the perceptual score defined in Eq. (1) is unambiguous.

Formally, to compute $\text{Acc}_{f,x,y}(\mathcal{M} \setminus \{M_m\})$ we don't use all modalities from data $x_i$. Instead, we use all modalities but $M_m$, *i.e.*, $x_i^{\mathcal{M} \setminus M_m}$ and append the data $x_j^{M_m}$ of modality $M_m$ from another data point $x_j$. Hereby $j$ is drawn uniformly from $\{1, \ldots, |\mathcal{D}|\}$, *i.e.*, from $\mathcal{U}(1, |\mathcal{D}|)$. Taken together we compute the accuracy via

$$\text{Acc}_{f,x,y}(\mathcal{M} \setminus \{M_m\}) = \mathbb{E}_{j \sim \mathcal{U}(1,|\mathcal{D}|)}[\mathbb{1}_{f(\{x^{\mathcal{M} \setminus M_m}, x_j^{M_m}\})=y}]. \tag{3}$$

In the following we discuss how to compute the normalization $Z$ employed in Eq. (1).

### 3.2.2 Normalization

Normalization enables comparability. For this, normalization aims for consistency of the perceptual score between models designed for the same task and between models designed for different tasks. Two types of normalization are useful to consider: 1) a *task-normalization*, which enables a more meaningful comparison of the perceptual score across different tasks; and 2) a *model-normalization* that enables a meaningful comparison of the perceptual score within the same task. We think models should be analyzed with both kinds of normalization in mind.

**Task-normalization:** Task normalization is particularly useful if a trivial classifier's accuracy is high. Without normalization, the comparison of models designed for different tasks is inconsistent. For instance, if a task is relatively easy, the use of permuted data won't modify the accuracy much because the task can be addressed with any data modality. This would result in a perceptual score close to zero, which isn't compelling because, in this case, even a marginal reduction in performance might be significant.

The majority vote classifier always predicts the majority class of the employed training set. We use $\widehat{\text{Acc}}_{\mathcal{D}}$ to refer to the accuracy of the majority vote classifier evaluated on the test set $\mathcal{D}$ and compute the normalization factor via

$$Z_{\mathcal{D}} = 1 - \widehat{\text{Acc}}_{\mathcal{D}}. \tag{4}$$

Intuitively, the normalization factor is the gap between the perfect accuracy (*i.e.*, 1) and the accuracy of the majority vote classifier, *i.e.*, $\widehat{\text{Acc}}_{\mathcal{D}}$. Hence, normalization by the majority vote accuracy

amplifies the score difference as we compare the obtained gain (classifier accuracy minus permuted classifier accuracy) to the maximally possible gain (one minus majority vote classifier accuracy).

We note that this normalization may result in perceptual score higher than one since the majority vote may be superior to the permuted accuracy. However, this case is unlikely in practice and did not occur in any of our experiments. Notably, this normalization is limited by the fact that the model accuracy is not considered, as we will discuss next.

**Model-normalization:** The performance of a model is an important factor. Suppose for two different classifiers permutation of a modality results in the same accuracy as that of the majority vote classifier. As a result, the initially stronger model will attain a higher perceptual score. This property can be desirable since one may want the perceptual score to reflect both perception and the ability to address a task. However, in this case the perceptual score does not solely reflect the degree to which a model considers a particular modality for decision making. Instead the perceptual score for a modality would be conflated with the model's accuracy.

To obtain a score which solely reflects the degree to which a model considers a particular modality, we normalize by the model's accuracy. Formally, we normalize via

$$Z_{\mathcal{D},f} = \mathbb{E}_{(x,y)\sim\mathcal{D}}[\mathrm{Acc}_{f,x,y}(\mathcal{M})]. \tag{5}$$

# 4 Evaluation of Perceptual Scores

In the following, we assess the perceptual score using popular multi-modal datasets. Specifically, in Sec. 4.1, we study visual question answering (VQA, VQA-CP). We examine video social reasoning (*i.e.*, SocialIQ) in Sec. 4.2. In Sec. 4.3, we assess visual dialog models. Our analysis shows that state-of-the-art models exploit biases that haven't been documented. We study the bias by investigating samples with low perceptual scores and discover its cause.

**Experimental setup:** We compute the perceptual score based on five permutations per sample. We calculate the perceptual score five times with different permutations and report the mean score along with the standard deviations. We find the expectation to converge quickly and to be stable. For all the models, we used the official implementations. For more details please see our implementation.[1]

## 4.1 Visual Question Answering

The visual question answering task reasons about an image given a question. We use the VQAv2 dataset [20], which contains 443,757 image-question pairs in the train set and 214,354 in the validation set. We also assess the perceptiveness of models trained on 'Visual Question Answering: Changing Priors' (VQA-CP) data [21], which was released after several studies suggested that VQAv2 models heavily rely on answer priors. For instance, 'how many' questions are typically answered with '2.' To overcome this shortcoming, VQA-CP suggested a new train-test-split. As a result, the train and test sets have different prior distributions for each question type. The new split consists of 438,183 training samples, and 219,928 samples for validation.

**Baselines:** We use four baselines for VQAv2: 1) BUTD [49], an early competitive approach that used detector-based features pre-trained on VisualGenome [50]; 2) BAN [51], a baseline that uses an effective multi-modal bilinear attention; 3) LMH [41], originally crafted for VQA-CP, this approach removes superficial question patterns; and 4) LXMERT [52] a large-scale Transformer-based model that is pre-trained with large amounts of image-text pairs. For VQA-CP, we also analyze: 1) CSS [53] which generates counterfactual training samples by masking critical objects in images or words in questions; 2) RMFE [42] which maximizes the amount of information that each modality contributes to the prediction.

### 4.1.1 Quantitative Analysis

In Tab. 1, we provide the perceptual score and the accuracy for different baselines and questions. We start by analyzing the perceptual scores of the vision and language modalities. In most cases, models perceive the question better (*i.e.*, $P_Q > P_V$). Studying the accuracy, LXMERT has the

---

[1]https://github.com/itaigat/perceptual-score

Table 1: Accuracy and perceptual scores on VQAv2 and VQA-CP for different baselines and question types: number (Num), yes/no (Y/N), and other. We report the accuracy ($\text{Acc}_\mathcal{M}$), the accuracy after removing a modality's influence ($\text{Acc}_{\mathcal{M}\setminus\{V\}}$, $\text{Acc}_{\mathcal{M}\setminus\{Q\}}$), the perceptual score without normalization ($P_V$, $P_Q$), the perceptual score with task normalization ($P_V/Z_\mathcal{D}$, $P_Q/Z_\mathcal{D}$), the perceptual score with model normalization ($P_Q/Z_{\mathcal{D},f}$, $P_V/Z_{\mathcal{D},f}$), and majority-vote accuracy ($\widehat{\text{Acc}_\mathcal{D}}$). Means and standard deviations are provided.

| Model | Q. Type | $\text{Acc}_\mathcal{M}$ | Image | | | | Question | | | | $\widehat{\text{Acc}_\mathcal{D}}$ |
|---|---|---|---|---|---|---|---|---|---|---|---|
| | | | $\text{Acc}_{\mathcal{M}\setminus\{V\}}$ | $P_V$ | $P_V/Z_\mathcal{D}$ | $P_V/Z_{\mathcal{D},f}$ | $\text{Acc}_{\mathcal{M}\setminus\{Q\}}$ | $P_Q$ | $P_Q/Z_\mathcal{D}$ | $P_Q/Z_{\mathcal{D},f}$ | |
| | | | | | VQAv2 | | | | | | |
| LXMERT | All | 68.97 | 36.46 | 32.51 | $47.40 \pm 0.40$ | $47.16 \pm 0.37$ | 27.41 | 41.56 | $60.60 \pm 0.38$ | $60.26 \pm 0.38$ | 31.42 |
| LMH | All | 54.33 | 27.90 | 26.43 | $38.54 \pm 0.32$ | $48.64 \pm 0.29$ | 22.82 | 31.51 | $45.94 \pm 0.30$ | $57.99 \pm 0.31$ | 31.42 |
| BAN | All | 65.67 | 35.09 | 30.58 | $44.59 \pm 0.38$ | $46.56 \pm 0.39$ | 28.25 | 37.43 | $54.57 \pm 0.36$ | $56.99 \pm 0.32$ | 31.42 |
| BUTD | All | 63.09 | 34.38 | 28.71 | $41.86 \pm 0.36$ | $45.52 \pm 0.38$ | 28.78 | 34.31 | $50.03 \pm 0.35$ | $54.33 \pm 0.34$ | 31.42 |
| LXMERT | Num | 52.73 | 14.55 | 38.18 | $47.14 \pm 0.28$ | $72.40 \pm 0.00$ | 13.04 | 39.69 | $49.01 \pm 0.25$ | $49.00 \pm 0.25$ | 19.01 |
| LMH | Num | 37.58 | 15.64 | 21.94 | $27.09 \pm 0.26$ | $58.38 \pm 0.26$ | 13.75 | 23.82 | $29.41 \pm 0.22$ | $63.40 \pm 0.27$ | 19.01 |
| BAN | Num | 48.62 | 18.55 | 30.07 | $37.13 \pm 0.28$ | $61.84 \pm 0.29$ | 16.07 | 32.55 | $40.19 \pm 0.24$ | $66.95 \pm 0.27$ | 19.01 |
| BUTD | Num | 42.46 | 21.04 | 21.41 | $26.44 \pm 0.25$ | $50.47 \pm 0.24$ | 18.39 | 24.06 | $29.71 \pm 0.22$ | $56.29 \pm 0.26$ | 19.01 |
| LXMERT | Other | 60.86 | 16.81 | 44.05 | $45.63 \pm 0.27$ | $72.37 \pm 0.27$ | 3.47 | 57.39 | $59.45 \pm 0.08$ | $94.30 \pm 0.09$ | 3.47 |
| LMH | Other | 54.29 | 13.72 | 40.58 | $42.04 \pm 0.21$ | $74.73 \pm 0.21$ | 4.30 | 49.99 | $51.79 \pm 0.09$ | $92.08 \pm 0.08$ | 3.47 |
| BAN | Other | 56.99 | 16.37 | 40.62 | $42.08 \pm 0.25$ | $71.27 \pm 0.25$ | 4.17 | 52.82 | $54.72 \pm 0.10$ | $92.68 \pm 0.06$ | 3.47 |
| BUTD | Other | 54.99 | 14.29 | 40.70 | $42.16 \pm 0.22$ | $73.93 \pm 0.22$ | 4.53 | 50.46 | $52.27 \pm 0.09$ | $91.74 \pm 0.07$ | 3.47 |
| LXMERT | Y/N | 85.30 | 64.58 | 20.72 | $41.02 \pm 0.40$ | $24.29 \pm 0.32$ | 63.84 | 21.46 | $42.49 \pm 0.32$ | $25.16 \pm 0.38$ | 49.49 |
| LMH | Y/N | 60.24 | 50.80 | 9.43 | $18.68 \pm 0.33$ | $15.66 \pm 0.42$ | 50.29 | 9.94 | $19.68 \pm 0.29$ | $16.50 \pm 0.39$ | 49.49 |
| BAN | Y/N | 83.03 | 65.44 | 17.59 | $34.82 \pm 0.36$ | $21.18 \pm 0.29$ | 64.09 | 18.93 | $37.48 \pm 0.33$ | $22.80 \pm 0.32$ | 49.49 |
| BUTD | Y/N | 80.94 | 65.42 | 15.52 | $30.73 \pm 0.32$ | $19.27 \pm 0.33$ | 64.24 | 16.69 | $33.05 \pm 0.30$ | $20.63 \pm 0.25$ | 49.49 |
| | | | | | VQA-CP | | | | | | |
| CSS | All | 57.89 | 36.46 | 21.43 | $26.43 \pm 0.41$ | $37.01 \pm 0.27$ | 7.93 | 49.96 | $61.61 \pm 0.21$ | $86.30 \pm 0.31$ | 18.91 |
| RMFE | All | 54.20 | 29.91 | 24.29 | $27.11 \pm 0.34$ | $47.17 \pm 0.31$ | 7.65 | 46.55 | $51.95 \pm 0.17$ | $55.91 \pm 0.24$ | 10.40 |
| LMH | All | 52.23 | 27.14 | 25.09 | $28.00 \pm 0.32$ | $45.35 \pm 0.24$ | 7.13 | 45.10 | $50.33 \pm 0.14$ | $54.08 \pm 0.27$ | 10.40 |
| CSS | Num | 51.34 | 44.50 | 6.84 | $7.20 \pm 0.38$ | $13.32 \pm 0.08$ | 13.04 | 38.30 | $40.34 \pm 0.29$ | $74.60 \pm 0.31$ | 5.06 |
| RMFE | Num | 44.03 | 37.25 | 6.77 | $7.13 \pm 0.33$ | $18.84 \pm 0.30$ | 34.05 | 9.97 | $10.51 \pm 0.30$ | $26.51 \pm 0.33$ | 5.06 |
| LMH | Num | 37.40 | 30.35 | 7.05 | $7.43 \pm 0.28$ | $15.39 \pm 0.28$ | 27.48 | 9.92 | $10.45 \pm 0.25$ | $22.65 \pm 0.28$ | 5.06 |
| CSS | Other | 46.48 | 10.12 | 36.36 | $37.57 \pm 0.15$ | $78.22 \pm 0.15$ | 4.12 | 42.36 | $43.77 \pm 0.16$ | $91.13 \pm 0.14$ | 3.23 |
| RMFE | Other | 45.97 | 10.08 | 35.89 | $37.09 \pm 0.17$ | $77.94 \pm 0.18$ | 4.15 | 41.82 | $43.22 \pm 0.11$ | $91.14 \pm 0.21$ | 3.23 |
| LMH | Other | 46.10 | 10.17 | 35.93 | $37.13 \pm 0.18$ | $78.07 \pm 0.19$ | 4.08 | 42.02 | $43.42 \pm 0.11$ | $90.96 \pm 0.23$ | 3.23 |
| CSS | Yes/No | 83.11 | 82.52 | 0.59 | $0.71 \pm 0.04$ | $3.16 \pm 0.04$ | 43.84 | 39.27 | $100.0 \pm 0.00$ | $47.25 \pm 0.33$ | 64.46 |
| RMFE | Yes/No | 74.47 | 62.49 | 11.98 | $18.58 \pm 0.31$ | $17.99 \pm 0.31$ | 59.31 | 15.16 | $23.51 \pm 0.28$ | $21.68 \pm 0.31$ | 35.52 |
| LMH | Yes/No | 73.75 | 60.49 | 13.27 | $20.58 \pm 0.32$ | $16.08 \pm 0.32$ | 57.76 | 16.00 | $24.81 \pm 0.29$ | $20.35 \pm 0.32$ | 35.52 |

highest accuracy (68.97%). In contrast, LMH, which reduces the reliance on question priors, achieves the lowest accuracy (54.33%). However, the model-normalized perceptual score for the visual modality ($P_V/Z_{\mathcal{D},f}$) suggests: LMH perceives image data similarly to other models. Studying the task normalized perceptual score for the visual modality ($P_V/Z_\mathcal{D}$) suggests: despite the high accuracy on 'Y/N' questions, their image and question perceptual scores are very low, *i.e.*, models mostly ignore the visual data and rely on priors.

Next, we show metrics for VQA-CP, a variant designed to reduce bias caused by answer priors. We compare different models on all the data by analyzing the model-normalized perceptual score for both visual and question modalities ($P_V/Z_{\mathcal{D},f}$ and $P_Q/Z_{\mathcal{D},f}$). Interestingly, the state-of-the-art model, CSS, has the lowest image perceptual score (37.01%) and the highest question perceptual score (86.30%), suggesting that the question modality may serve as a shortcut to answer without perceiving the image. Further, by analyzing the different question types via the task-normalized score ($P_V/Z_\mathcal{D}$), we note that CSS has a perceptual score for the image modality of only 3.6% for 'Y/N' questions. One possible explanation: CSS generates new samples, which in turn alter priors perceived during training. We further investigate this hypothesis next.

### 4.1.2 Bias Analysis for CSS

The Counterfactual Samples Synthesizing (CSS) model produces counterfactual training samples by masking either critical objects in images or words in questions, and by assigning different ground-truth answers. Our perceptiveness study above shows that the CSS model has a significantly lower perceptual score for the visual modality, despite being state-of-the-art on VQA-CP with a substantial accuracy gap of 6.5% over LMH. Why?

Table 2: Proportion of yes/no ratios for different kinds of questions. The initial token categorizes questions. We report proportion in the train set, the test set, and the model prediction. '# Train' indicates the number of samples in the train set, '# Test' is the number of samples in the test set.

| Token | Model | Predicted Yes | Predicted No | Test Yes | Test No | Train Yes | Train No | # Test | # Train |
|-------|-------|---------------|--------------|----------|---------|-----------|----------|--------|---------|
| 'has' | CSS | 1.0 | 0.0 | 1.0 | 0.0 | 0.5 | 0.5 | 1414 | 2686 |
|       | LMH | 0.47 | 0.53 | 1.0 | 0.0 | 0.0 | 1.0 | 1414 | 1343 |
| 'can' | CSS | 0.94 | 0.06 | 0.72 | 0.28 | 0.5 | 0.5 | 3110 | 2546 |
|       | LMH | 0.38 | 0.62 | 0.72 | 0.28 | 0.0 | 1.0 | 3110 | 1273 |
| 'is a' | CSS | 0.01 | 0.99 | 0.0 | 1.0 | 0.5 | 0.5 | 324 | 654 |
|        | LMH | 0.37 | 0.63 | 0.0 | 1.0 | 1.0 | 0.0 | 324 | 327 |
| 'do' | CSS | 0.99 | 0.01 | 1.0 | 0.0 | 0.5 | 0.5 | 1388 | 11302 |
|      | LMH | 0.42 | 0.58 | 1.0 | 0.0 | 0.44 | 0.56 | 1388 | 5651 |

The first point to note: CSS generates new samples which may shift prior distributions leveraged during training. Recalling that VQA-CP was introduced to prevent benefiting from answer priors in VQAv2 reveals a potential reason for the improvements of CSS.

Use of the sample perceptual score permits a more in-depth analysis. In Tab. 2, we identify popular question start tokens with low sample perceptual scores for the visual modality ('do,' 'has,' 'can,' 'is a'). We further examine their prediction accuracy using both CSS and LMH. We find that the proportion of 'yes' and 'no' answers between the train and the test set differ: the yes answer is correct for 88% of the questions in the test set starting with 'has,' while the yes answer is *never* correct for the corresponding training set questions. For CSS, counterfactual samples, however, produce equal proportions. As a result, CSS seemingly alleviates the prior inconsistency and adjusts the majority of its predictions to 'yes,' which more closely resembles the test set.

Use of the perceptual scores hence permits to hypothesize: improvements in CSS can be attributed to a shifted prior distribution instead of a complex counterfactual data-manipulation. To test this hypothesis we train the LMH model using samples obtained from CSS training. Importantly, we did not modify the image or the question. *Without even any changes to the input*, we obtain an accuracy of 57.54%, only 0.3% lower than CSS and within standard deviation.

## 4.2 Video Social Reasoning

SocialIQ [28] proposes an unconstrained benchmark, specifically designed to understand social situations. More concretely, given an input tuple of a video, a question, and an answer, the task is to predict whether the answer is correct or not. The videos were collected from YouTube and annotated by students. The dataset is split into 37,191 training samples, and 5,320 validation set samples.

**Baseline:** We were able to reproduce the SocialIQ baseline [28] and achieve an accuracy of 64.84% using all feature modalities during training. In fact, we achieved an accuracy of 67.38% by changing the baseline's model to FGA [54] and by using GloVe [55] instead of BERT features [56].

### 4.2.1 Quantitative Analysis

In Tab. 3 we show scores for different metrics. The task-normalized perceptual scores $(P_M/Z_D)$ for different modalities $M$ (first column) of FGA and the baseline reveals heavy reliance on the answer modality and low dependence on the video modality (*i.e.*, 15.24 *vs.* 3.25). To verify that the answer is indeed the only important modality, we train a classifier using *only* the answer modality and observe: we can surpass the original paper's baseline by 4% when *only* using answer features, achieving 68.65% accuracy. Again, the perceptual score helped us understand shortcomings of a model. Upon analyzing the bias source, we observe a sentiment bias, which we discuss next.

### 4.2.2 Bias Analysis for SocialIQ

Assessing samples with low sample perceptual scores can reveal biases. Our study suggests that the labels correlate strongly with the sentiment. In Fig. 3, we marked with a red border two videos with a low sample perceptual score for the video modality. Studying those and similar samples, we find:

Table 3: Accuracy and perceptual scores on SocialIQ. For different modalities (first column), we report the accuracy ($\text{Acc}_\mathcal{M}$), the accuracy after removing the modality's influence ($\text{Acc}_{\mathcal{M}\setminus\{M\}}$), the perceptual score without normalization ($P_M$), the perceptual score with task normalization ($P_M/Z_\mathcal{D}$), the perceptual score with model normalization ($P_M/Z_{\mathcal{D},f}$), and the majority-vote accuracy ($\widehat{\text{Acc}_\mathcal{D}}$).

| Modality $M$ | Model | $\text{Acc}_\mathcal{M}$ | $\text{Acc}_{\mathcal{M}\setminus\{M\}}$ | $P_M$ | $P_M/Z_\mathcal{D}$ | $P_M/Z_{\mathcal{D},f}$ | $\widehat{\text{Acc}_\mathcal{D}}$ |
|---|---|---|---|---|---|---|---|
| Answer | Baseline | 64.84 | 56.73 | 8.11 | $18.92 \pm 0.13$ | $12.51 \pm 0.03$ | 57.14 |
| | FGA | 67.38 | 57.11 | 10.27 | $23.96 \pm 0.21$ | $15.24 \pm 0.10$ | 57.14 |
| Question | Baseline | 64.84 | 64.32 | 0.52 | $1.21 \pm 0.02$ | $0.80 \pm 0.03$ | 57.14 |
| | FGA | 67.38 | 65.19 | 2.19 | $5.11 \pm 0.11$ | $3.25 \pm 0.08$ | 57.14 |
| Video | Baseline | 64.84 | 63.79 | 1.05 | $2.45 \pm 0.05$ | $1.62 \pm 0.03$ | 57.14 |
| | FGA | 67.38 | 64.42 | 2.96 | $6.91 \pm 0.16$ | $4.39 \pm 0.15$ | 57.14 |
| - | NLTK-Sentiment | 66.70 | 56.14 | 11.18 | $26.08 \pm 0.19$ | $16.36 \pm 0.09$ | 57.14 |
| - | Answer-Only | 68.65 | 57.29 | 11.39 | $26.57 \pm 0.21$ | $16.59 \pm 0.06$ | 57.14 |

Figure 3: SocialIQ data samples. On the left, we show a sample with a high perceptual score towards video data. Neither a positive nor a negative sentiment is evident in this sample. Hence, the video is required for prediction. We illustrate two samples (marked with a red border) that received a low perceptual score. There is a sentiment-based correlation between the label and the answer in these samples. For simplicity, we highlight with red color words that exhibit sentiment.

1) when the answer is True, the answer has a positive sentiment; 2) in contrast, when the answer is False, the answer contains words with a negative connotation, *e.g.*, 'uncomfortable.'

We hypothesize: successful prediction of the answer by just looking at the answer modality for SocialIQ data is due to sentiment-biased annotations. To validate this hypothesis, we use an off-the-shelf sentiment classifier from the NLTK package [57]. When applied to SocialIQ *without any training*, we obtain a remarkable answer prediction accuracy of 66.7%. This matches our reported result of 68.65% quite reasonably and outperforms the SocialIQ baseline [28].

## 4.3 Visual Dialog

The visual dialog task encourages models to ask *and* answer questions about visual input. Notably, each dialog-interaction employs many modalities (*e.g.*, image, question, caption, dialog-history). We show our results on the VisDial v1.0 dataset, where 123,287 images are used for training, 2,000 images for validation, and 8,000 images for testing [58]. Each image is associated with ten questions, and each question has 100 corresponding answer candidates.

Instead of accuracy $\text{Acc}_{f,x,y}(\mathcal{M})$, here, we use ranking-based metrics as the Visual Dialog dataset primarily uses two metrics: MRR and NDCG. In short, the MRR metric examines the rank of a single ground-truth response (*i.e.*, sparse annotations), while the NDCG metric measures the cumulative gain in the case of multiple correct answers (*i.e.*, dense annotations). We computed the majority ranking based on answer frequency in the train set. See appendix for more.

**Baselines:** We use two baselines for VisDial v1.0: 1) FGA [54], an attention unit inspired from graphical models that infers an attention map for each modality; and 2) LS [59], which pre-trains on related vision-language datasets, *e.g.*, Conceptual Captions and Visual Question Answering [1, 60]. We also report LS (CE), which finetunes on the dense annotations, at the expense of MRR performance.

Table 4: Perceptual scores on VisDial v1.0. We show that the perceptual score can be computed using metrics other than accuracy (*i.e.*, MRR and NDCG).

| Modality $M$ | Model | MRR | | | | | | NDCG | | | | | |
|---|---|---|---|---|---|---|---|---|---|---|---|---|---|
| | | $MRR_{\mathcal{M}}$ | $MRR_{\mathcal{M}\setminus\{M\}}$ | $P_M$ | $P_M/Z_{\mathcal{D}}$ | $P_M/Z_{\mathcal{D},f}$ | $\widehat{MRR}_{\mathcal{D}}$ | $NDCG_{\mathcal{M}}$ | $NDCG_{\mathcal{M}\setminus\{M\}}$ | $P_M$ | $P_M/Z_{\mathcal{D}}$ | $P_M/Z_{\mathcal{D},f}$ | $\widehat{NDCG}_{\mathcal{D}}$ |
| Question | LS (CE) | 52.21 | 32.48 | 19.73 | $29.11 \pm 0.02$ | $37.79 \pm 0.03$ | 32.22 | 75.24 | 20.35 | 54.89 | $73.86 \pm 0.04$ | $72.95 \pm 0.03$ | 25.68 |
| | LS | 69.00 | 47.96 | 21.04 | $31.04 \pm 0.03$ | $30.49 \pm 0.03$ | 32.22 | 64.89 | 23.92 | 40.97 | $55.13 \pm 0.02$ | $63.14 \pm 0.01$ | 25.68 |
| | FGA | 66.14 | 32.23 | 33.91 | $50.03 \pm 0.02$ | $51.26 \pm 0.03$ | 32.22 | 56.00 | 30.74 | 25.26 | $33.99 \pm 0.02$ | $45.11 \pm 0.02$ | 25.68 |
| Image | LS (CE) | 52.21 | 34.94 | 17.27 | $25.48 \pm 0.02$ | $33.08 \pm 0.02$ | 32.22 | 75.24 | 57.45 | 17.79 | $23.94 \pm 0.02$ | $23.64 \pm 0.03$ | 25.68 |
| | LS | 69.00 | 52.06 | 16.94 | $24.99 \pm 0.01$ | $24.55 \pm 0.03$ | 32.22 | 64.89 | 51.50 | 13.39 | $18.02 \pm 0.01$ | $20.63 \pm 0.03$ | 25.68 |
| | FGA | 66.14 | 52.15 | 14.00 | $20.65 \pm 0.01$ | $21.16 \pm 0.02$ | 32.22 | 56.00 | 46.73 | 9.27 | $9.27 \pm 0.03$ | $16.55 \pm 0.02$ | 25.68 |
| Caption | LS (CE) | 52.21 | 52.18 | 0.03 | $0.00 \pm 0.00$ | $0.06 \pm 0.00$ | 32.22 | 75.24 | 75.19 | 0.05 | $0.07 \pm 0.00$ | $0.07 \pm 0.01$ | 25.68 |
| | LS | 69.00 | 69.00 | 0.00 | $0.00 \pm 0.00$ | $0.00 \pm 0.00$ | 32.22 | 64.89 | 64.89 | 0.00 | $0.00 \pm 0.00$ | $0.00 \pm 0.00$ | 25.68 |
| | FGA | 66.14 | 64.93 | 1.21 | $1.79 \pm 0.02$ | $1.83 \pm 0.02$ | 32.22 | 56.00 | 55.32 | 0.68 | $0.92 \pm 0.01$ | $1.22 \pm 0.01$ | 25.68 |

Table 5: Perceptual scores for audio-visual crowd counting on high and low resolution settings.

| Image quality | $1 - MAPE_{\mathcal{M}}$ | $1 - MAPE_{\mathcal{M}\setminus A}$ | $P_A$ | $P_A \setminus Z_{\mathcal{D}}$ | $P_A \setminus Z_{\mathcal{D},f}$ | $1 - MAPE_{\mathcal{M}\setminus V}$ | $P_V$ | $P_V \setminus Z_{\mathcal{D}}$ | $P_V \setminus Z_{\mathcal{D},f}$ | $1 - \widehat{MAPE}_{\mathcal{D}}$ |
|---|---|---|---|---|---|---|---|---|---|---|
| High | 77.71 | 73.88 | 3.83 | 9.22 | 4.93 | 43.16 | 34.55 | 83.21 | 44.46 | 58.48 |
| Low | 72.51 | 66.58 | 9.93 | 23.91 | 13.69 | 45.07 | 27.44 | 66.08 | 37.84 | 58.48 |

#### 4.3.1 Quantitative Analysis

In Tab. 4, we show the perceptual scores for different modalities (*i.e.*, $M$), and for two metrics: MRR and NDCG. We find: 1) using the MRR metric, the model-normalized perceptual score for the question modality of FGA is relatively high compared to LS (*i.e.*, 51.26% *vs.* 30.49%). However, when we employ the NDCG metric, the LS model perceives the question better (*i.e.*, 63.14% *vs.* 45.11%). 2) analyzing the only model optimized for NDCG, *i.e.*, LS (CE), reveals: When analyzed using the NDCG metric, the model-normalized perceptual score for both image and question is significantly higher than for MRR. Interestingly, the LS (CE) model-normalized perceptual score for the image modality is on par with the other models when we use the MRR metric. However, the question perceptual score is low, which suggests the reason for the low MRR performance may be related to the low utilization of the question. Finally, we note that all models, on both metrics, seem to ignore the caption, which suggests caption information is redundant.

### 4.4 Audio-visual Crowd Counting

We further investigate a model for audio-visual crowd counting task. Given an image and an utterance, the task is to count the number of people in a scene. We use the auDiovISual Crowd cOunting (DISCO) dataset, consisting of 1,935 images and the corresponding audio clips, and 170,270 annotated instances. Hu et al. [61] hypothesize that in extreme cases (*i.e.*, low-quality images), the scene audio can improve the prediction. As expected, the perceptual score is able to quantify the degree to which each modality is perceived. In Tab. 5 we compare the low-quality and high-quality images by employing task normalization. It reveals that in high-quality image settings, the audio modality is mostly ignored (the task-normalized audio perceptual score is 9.22%). In contrast, the audio modality perceptiveness for low-quality image settings is significant (the task-normalized audio perceptual score is 23.91%).

## 5 Conclusion

We introduced the perceptual score of a multi-modal classifier towards a data modality. The perceptual score assesses a classifier's perceptiveness of a modality and reveals exciting insights if analyzed carefully. We hope that this is demonstrated by our studies which reveal that 1) shifted prior distributions seem to help CSS achieve state-of-the-art results, 2) SocialIQ data exhibits a sentiment bias, 3) Visual Dialog caption information appears less informative, and 4) Audio perceptiveness for audio-visual crowd counting increases when low-quality images were present. We hope that researchers working on multi-modal models see the use of the perceptual score and will start to report the perceptual score in addition to classical accuracy metrics.

*Limitations:* We propose perceptual scores, a novel metric that conveys information about multi-modal classifiers. The two limitations we can see: 1) a small computational overhead; 2) a false sense of security. Perceptual scores don't alleviate the need to carefully study results. However we think the societal and scientific benefits of reporting this novel metric outweigh the concerns.

*Acknowledgements:* This work is supported in part by NSF Grant #1718221, 2008387, 2045586, 2106825, MRI #1725729, and NIFA 2020-67021-32799. We also acknowledge support from Samsung, Amazon, and Cisco Systems Inc. (Gift Award CG 1377144/thanks for access to Arcetri).

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
