# Appendix – Perceptual Score: What Data Modalities Does Your Model Perceive?

In this supplementary material, we provide a study on synthetic data (Sec. A), discussions on the Social IQ (Sec. 4.2 in the main paper), VisDial (Sec. 4.3 in the main paper), and Audio-visual Crowd Counting (Sec 4.4 in the main paper) datasets (Sec. B), and additional qualitative results.

## A   Synthetic data

We study the perceptual score with synthetic data consisting of three modalities $\{a, b, c\}$. This synthetic data permits control of every aspect of the data.

Specifically, we show that the perceptual score can identify two properties: (1) utilization of the modality; and (2) informativeness of the modality. In both cases, a low perceptual score is expected if the model does not use a modality or provide sufficient information to explain the label. Additionally, if two modalities explain the label equally, their perceptual score should be similar.

To study this, we generate synthetic data that involves non-linear interactions:

1. Sample $A \in \mathbb{R}^{d_1}, B \in \mathbb{R}^{d_2}, C \in \mathbb{R}^{d_3}$ from $U(-\tau, \tau)$.
2. Sample $\hat{a}, \hat{b} \in \mathbb{R} \sim N(0, 1)$ subject to $|\hat{a} \cdot \hat{b} + c| > \delta$.
3. Sample $\hat{c} \in \mathbb{R} \sim N(0, \sigma_c)$ where $\sigma_c \in \mathbb{R}$.
4. If $(\hat{a} \cdot \hat{b}) > 0$ then label $y = 1$ otherwise $y = 0$.
5. Return the data point $(\hat{a}A, \hat{b}B, \hat{c}C, y)$.

In order to compute a perceptual score, we use 20 permutations. We calculate the score ten times (*i.e.*, using 20 different permutations) and provide its standard deviation and mean. We create ten different datasets by varying the variance of modality $c$. We consider a logistic regression (*i.e.*, a linear model) and a two-layer neural network with a non-linear ReLU activation. Note, the logistic regression cannot model multiplicative non-linear interactions. Each dataset includes 2k samples, 1k for training, and 1k for testing. We use the following hyperparameters: $d_1 = 2000, d_2 = 1000, d_3 = 100, \delta = 0.25, \tau = 1$.

### A.1   Utilization

In Tab 1, we show that in a linear model, $a$ and $b$ have a low perceptual scores $P_a$ and $P_b$ ($P_a = 0.39$ and $P_b = 0.56$ on average). As modeling of $a$ and $b$ requires non-linearity, this is expected. In contrast, the perceptual score $P_a$ and $P_b$ is high in non-linear neural network (See Tab. 2; $P_a = 35.72, P_b = 36.8$ on average).

### A.2   Informativeness

To examine that the perceptual score reflects the informativeness of a modality, we control the variance of modality $c$ by increasing it from zero to one. As expected, as the variance of $c$ increases, the modality becomes more informative (the label depends directly on $c$), and the perceptual score of $c$ increases for both the neural network and the logistic regression. For example, the perceptual score is zero when the variance of $c$ is zero, and $P_c = 32.25$ when $c$'s variance is one.

35th Conference on Neural Information Processing Systems (NeurIPS 2021).

Table 1: Perceptual scores for logistic regression on the synthetic data described in Sec. A.

| Var(c) | $\text{Acc}_{\mathcal{M}}$ | $P_a$ | $P_a \setminus Z_{\mathcal{D}}$ | $P_a \setminus Z_{\mathcal{D},f}$ | $P_b$ | $P_b \setminus Z_{\mathcal{D}}$ | $P_b \setminus Z_{\mathcal{D},f}$ | $P_c$ | $P_c \setminus Z_{\mathcal{D}}$ | $P_c \setminus Z_{\mathcal{D},f}$ | $\widehat{\text{Acc}_{\mathcal{D}}}$ |
|---|---|---|---|---|---|---|---|---|---|---|---|
| 0 | 48 | -0.9±0.18 | -1.86 | -1.87 | -2.04±0.3 | -4.22 | -4.24 | 0±0 | 0 | 0 | 48.3 |
| 0.1 | 50.1 | -1.39±0.31 | -2.76 | -2.78 | 1.14±0.29 | 2.26 | 2.28 | 0.65±0.03 | 1.29 | 1.3 | 50.4 |
| 0.2 | 55.8 | -0.08±0.23 | -0.16 | -0.15 | 0.59±0.26 | 1.17 | 1.06 | 5.86±0.18 | 11.58 | 10.5 | 50.6 |
| 0.3 | 63.2 | 3.65±0.1 | 7.39 | 5.78 | 1.59±0.28 | 3.22 | 2.51 | 13.9±0.21 | 28.13 | 21.99 | 49.4 |
| 0.4 | 66 | 1.34±0.14 | 2.66 | 2.03 | 1.29±0.26 | 2.57 | 1.95 | 16.16±0.27 | 32.19 | 24.49 | 50.2 |
| 0.5 | 72.3 | 0.61±0.25 | 1.29 | 0.85 | 1.36±0.16 | 2.87 | 1.88 | 22.12±0.36 | 46.56 | 30.59 | 47.5 |
| 0.6 | 76.5 | 0.48±0.2 | 0.97 | 0.63 | 1.02±0.23 | 2.06 | 1.33 | 27.57±0.26 | 55.81 | 36.04 | 49.4 |
| 0.7 | 76.6 | 0.74±0.12 | 1.43 | 0.96 | 0.93±0.15 | 1.81 | 1.22 | 26.34±0.3 | 51.14 | 34.38 | 51.5 |
| 0.8 | 79.7 | 0.47±0.2 | 0.92 | 0.59 | 0.58±0.16 | 1.14 | 0.73 | 29.94±0.23 | 58.58 | 37.56 | 51.1 |
| 0.9 | 79.6 | -0.8±0.12 | -1.68 | -1.01 | -0.76±0.15 | -1.59 | -0.95 | 30±0.33 | 63.15 | 37.68 | 47.5 |
| 1 | 81.9 | 0.27±0.1 | 0.51 | 0.32 | 0.53±0.15 | 1.01 | 0.64 | 31.54±0.35 | 60.41 | 38.5 | 52.2 |

Table 2: Perceptual scores for non-linear neural network on the synthetic data described in Sec. A.

| Var(c) | $\text{Acc}_{\mathcal{M}}$ | $P_a$ | $P_a \setminus Z_{\mathcal{D}}$ | $P_a \setminus Z_{\mathcal{D},f}$ | $P_b$ | $P_b \setminus Z_{\mathcal{D}}$ | $P_b \setminus Z_{\mathcal{D},f}$ | $P_c$ | $P_c \setminus Z_{\mathcal{D}}$ | $P_c \setminus Z_{\mathcal{D},f}$ | $\widehat{\text{Acc}_{\mathcal{D}}}$ |
|---|---|---|---|---|---|---|---|---|---|---|---|
| 0 | 100 | 49.85±0.42 | 98.9 | 49.85 | 50.1±0.3 | 99.4 | 50.1 | 0±0 | 0 | 0 | 50.4 |
| 0.1 | 99.9 | 49.98±0.17 | 104.12 | 50.03 | 50.02±0.38 | 104.21 | 50.07 | 0±0 | 0 | 0 | 48 |
| 0.2 | 98.7 | 47.3±0.27 | 96.14 | 47.92 | 47.28±0.22 | 96.09 | 47.9 | 1.54±0.05 | 3.13 | 1.56 | 49.2 |
| 0.3 | 96.2 | 41.59±0.59 | 80.61 | 43.24 | 42.62±0.4 | 82.61 | 44.31 | 6.24±0.15 | 12.09 | 6.48 | 51.6 |
| 0.4 | 96.1 | 38.21±0.36 | 81.13 | 39.77 | 37.95±0.28 | 80.58 | 39.49 | 11.41±0.13 | 24.23 | 11.88 | 47.1 |
| 0.5 | 95.1 | 33.32±0.3 | 61.48 | 35.04 | 33.63±0.28 | 62.05 | 35.36 | 17.47±0.21 | 32.24 | 18.37 | 54.2 |
| 0.6 | 96.7 | 31.54±0.33 | 60.88 | 32.61 | 31.18±0.23 | 60.2 | 32.25 | 22.14±0.09 | 42.74 | 22.9 | 51.8 |
| 0.7 | 95.1 | 28.34±0.37 | 56.34 | 29.8 | 28.63±0.3 | 56.93 | 30.11 | 22.61±0.24 | 44.95 | 23.77 | 50.3 |
| 0.8 | 94.6 | 26.5±0.26 | 52.37 | 28.01 | 26.55±0.26 | 52.47 | 28.07 | 25.93±0.29 | 51.24 | 27.41 | 50.6 |
| 0.9 | 95.3 | 23.92±0.4 | 50.15 | 25.1 | 22.73±0.27 | 47.65 | 23.85 | 29.08±0.2 | 60.97 | 30.52 | 47.7 |
| 1 | 96.7 | 22.47±0.21 | 44.86 | 23.24 | 21.84±0.3 | 43.6 | 22.59 | 32.58±0.23 | 65.04 | 33.7 | 50.1 |

Note that, the symmetry property holds, *e.g.*, $a$ and $b$ contribute equally, and the perceptual score is similar (within the margin of error).

Regarding normalization:

1. As expected, trends do not change.

2. In the neural network case, the model-normalization is half of the task-normalization. This is expected when the majority-vote accuracy is centered, as the task normalization considers the ideal gain (*i.e.*, $1 - \text{Acc}_{\mathcal{D}} = 50$). Indeed, this facilitates the comparison of tasks since different tasks have a different ideal gain. Accordingly, a perceptual score of 50% is equal to the ideal gain and is considered perfect based on task normalization. Model normalization, on the other hand, is more appropriate for weak models. However, in this example, the model achieves near-perfect accuracy. Consequently, the model-normalization doesn't affect the perceptual score.

## B Results

The following sections provide more results and details about each task.

**SocialIQ.** In Fig. 1 and Fig. 2 we illustrate additional samples with low and high image sample perceptual scores. In Fig. 1, we show samples with high scores. According to NLTK, all the answers that were assigned high scores have a neutral sentiment. In Fig. 2 we show samples with low scores. In this case, the answers often have an easily detectable sentiment. Also, we observe that answer-correctness correlates with the sentiment of answers. For instance, the question "What is conveyed by the posture of the two girls?" and for the proposed answer "They are hostile and unfriendly", *i.e.*, "hostile" and "unfriendly" have a negative sentiment. Indeed the ground-truth label indicates the answer is incorrect.

**Visual Dialog.** For visual dialog, two metrics are standard, MRR and NDCG. The MRR metric focuses on a single human-derived ground-truth answer. However, the metric ignores many correct candidate answers. Differently, the NDCG considers the rank of all the correct answers. The metric relies on dense annotation, where three annotators were asked to mark all the correct candidate answers. Formally, the MRR metric, *i.e.*, the inverse harmonic mean of the rank, is defined as: $\text{MRR}_{f,x,y} \triangleq \frac{1}{d} \sum_{i=1}^{d} \frac{1}{r_x}$, where $r_x$ is the rank of the correct response for the dialog question $x$. In

the case of dense annotations (*i.e.*, multiple correct responses), we use the discounted cumulative gain over the $K$ correct answers: $\mathrm{DCG}_{\mathrm{f,x,y}} \triangleq \sum_{i=1}^{K} \frac{s_i^x}{\log_2(i+1)}$, where $s_i$ is a binary score, for question $x$ representing the fraction of annotators that marked the candidate at position $i$ as correct. We normalize by the ideal $\mathrm{DCG}_K$ score ($\mathrm{IDCG}_K$), *i.e.*, $\mathrm{NDCG}_{\mathrm{f,x,y}} \triangleq \frac{\mathrm{DCG}_{\mathrm{f,x,y}}}{\mathrm{IDCG}_K}$.

It is evident from our experiments that for both the MRR and NDCG metrics, the perceptual question score is higher than the perceptual image score (NDCG: 54.89 vs. 17.79, 40.97 vs. 13.39, 25.26; MRR: 19.73 vs. 17.27, 21.04 vs. 16.94, 33.91 vs. 14.00). We analyze the samples with a low image perceptual score for the NDCG and MRR metrics. We find that for the NDCG metric, the questions with low perceptual scores are usually uncertain, *i.e.*, correct answers are 'not sure' or 'I don't know.' Thus, the different annotations did not help to perceive the image better, and the models relied on answer-priors for both annotations.

**Audio-visual Crowd Counting.** A regression algorithm is involved with the audio-visual crowd counting task, which estimates how many people are in a crowd. The task is initially evaluated by a residual error term that is not bound. On the other hand, the perceptual score is built based on accuracy, which is a score between 0 and 1. Accordingly, in order to ensure correctness, we emply the mean absolute percentage error (MAPE), *i.e.*, $\mathrm{MAPE}_{f,x,y} \triangleq \frac{100}{d} \sum_{i=1}^{d} \left| \frac{y_i - f(x_i)}{y_i} \right|$, where $y_i$ is the ground-truth value, $d$ is the test set size, and $f(x_i)$ is the model prediction.

Question: Why does the woman nod at 0:36?
Answer: The woman agrees with what the man has just said to her.
label: False

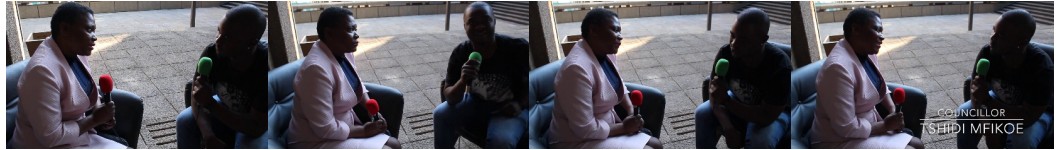

Question: Why does the girl with the sweater talk slowly when giving her answer?
Answer: she is unsure of her answer
label: True

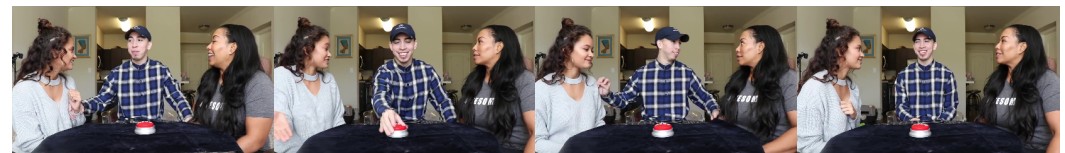

Question: Why is the man on the left yelling?
Answer: The man on the left is yelling because he feels that yelling will help him get his point across.
label: True

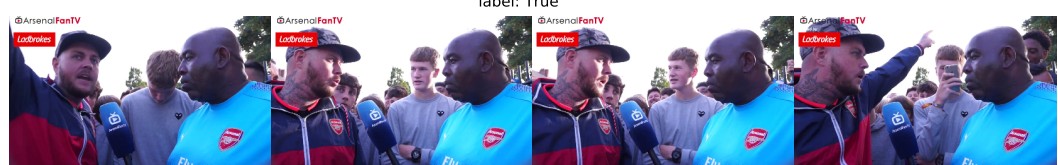

Question: How does the audience respond to his finale?
Answer: The audience responds with horror to the man's finale; the noise that they make reflects this horror
label: False

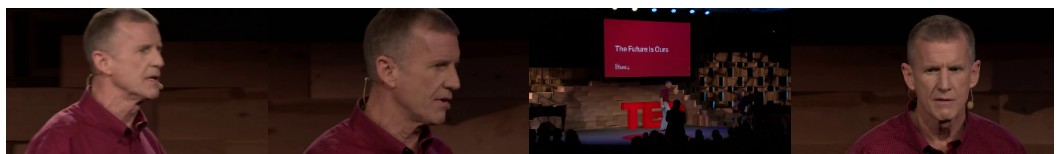

Question: Why does the man with the yellow hat talk about his personal life?
Answer: Because the man thinks that his personal life is very interesting
label: False

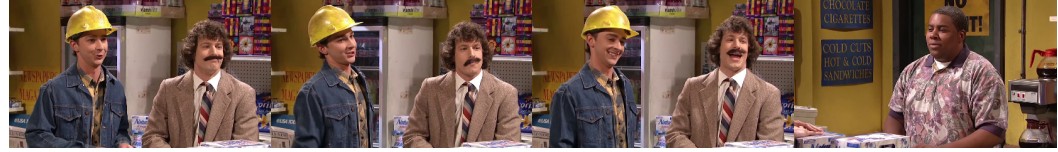

Question: What is the dynamic between the two individuals?
Answer: The two individuals are respectful of one another because they are scheduling a violent protest.
label: False

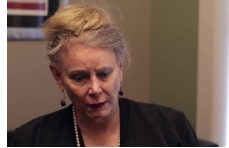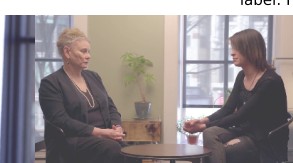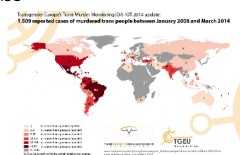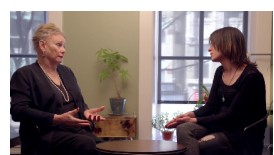

Question: What did the man think of the suggestion to use the 1000 dollars?
Answer: The man thinks the suggestion to use a thousand dollars a certain way should be amended because there is no thousand dollars.
label: True

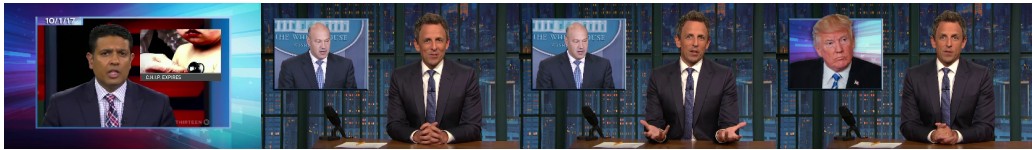

Question: Why does the man need to ask his family before receiving an invitation?
Answer: The man needs to ask his family before receiving an invitation because he needs their approval.
label: True

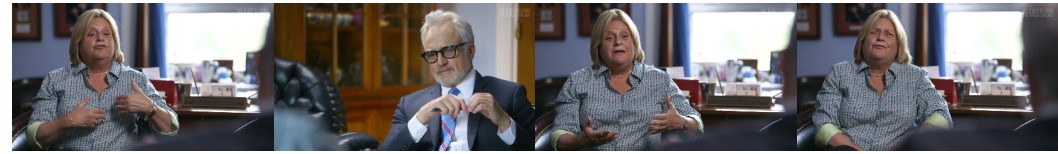

Question: How do the girls feel towards the game they are about to play?
Answer: They are dreading playing it
label: False

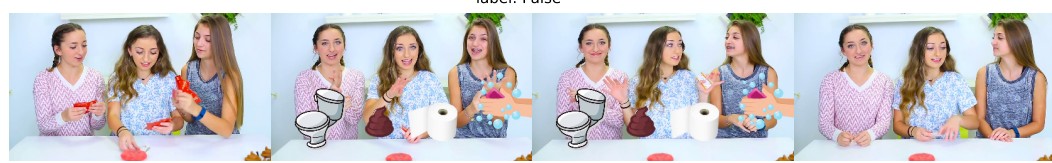

Question: What is the woman's tone?
Answer: The woman has a serious and formal tone in the video.
label: False

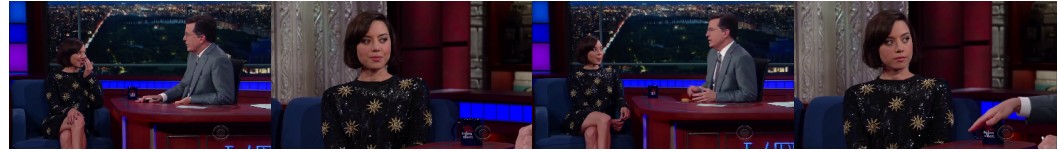

Question: How do they describe morning practices?
Answer: They do not like them.
label: True

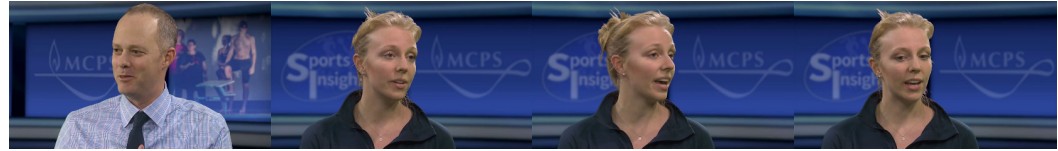

Question: Why is the man with glasses in the blue suit surprised?
Answer: He doesn't understand why the man was appointed.
label: True

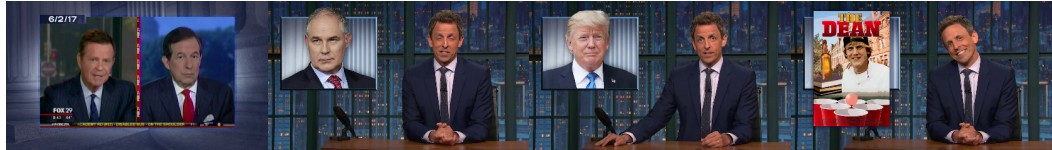

Question: Why does Ellis nod her head when Nadia is talking?
Answer: She agrees with everything Nadia believes
label: False

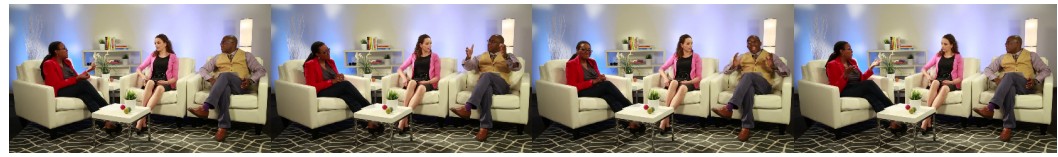

Question: How does the woman feel?
Answer: The woman feels very worried in the video.
label: True

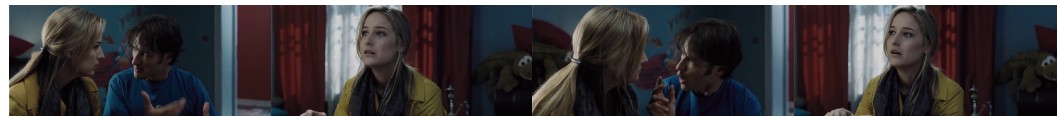

Question: Why does the woman in black congratulate the other women?
Answer: The woman in black congratulates the other women because she wants to convey the importance of their achievements.
label: True

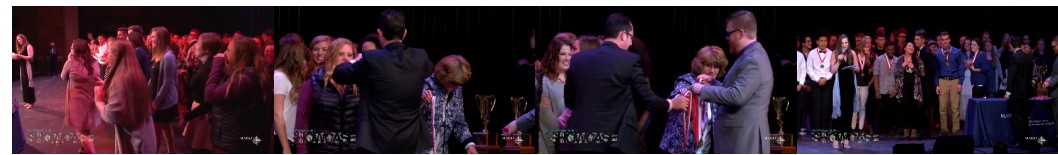

Question: How does the man on the right feel about the man on the left?
Answer: The man on the right thinks the man on the left is a fun conversationalist
label: True

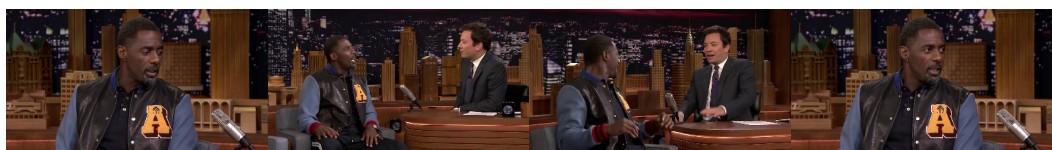

Question: Why does the man repeat what the woman says to him?
Answer: The man repeats what the woman says to him because he is uncertain of how to answer her question, and wants to buy time
label: True

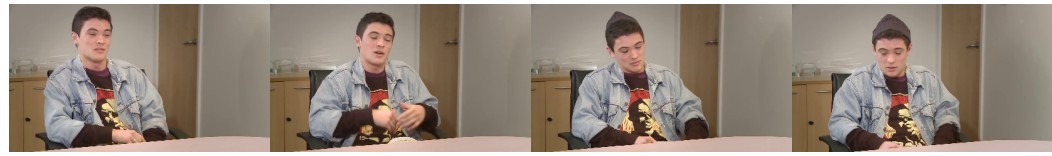

Question: Why is the woman so infuriated?
Answer: She is infuriated because she didn't eat.
label: False

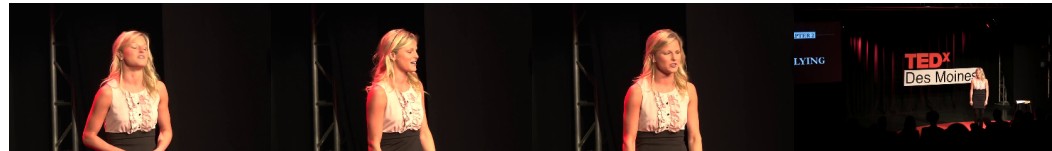

Question: Why does the woman in the red dress make an annoyed/bored face at the woman in the suit's comment about returning to the previous discussion topic?
Answer: She makes a face because she wants to talk about the previous discussion topic more
label: False

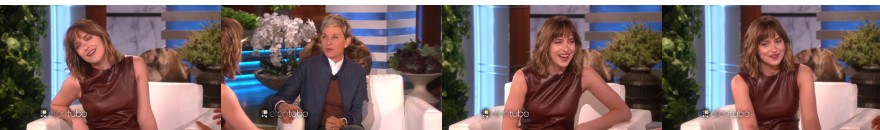

Question: What does the woman on the right feel about determining a law unconstitutional after it passes?
Answer: The woman on the right takes making a law unconstitutional very seriously.
label: True

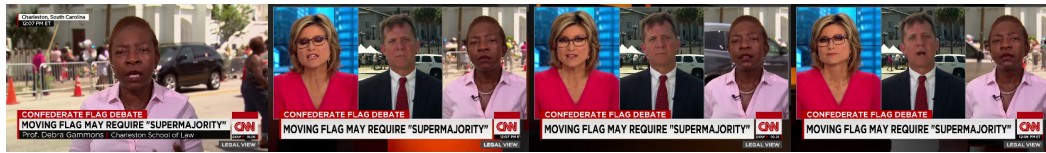

Question: What is the atmosphere like throughout the video?
Answer: The atmosphere throughout the video is very comical
label: True

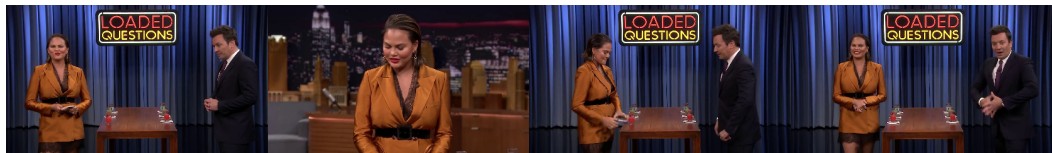

Question: Why is the man confused?
Answer: The man is confused because he thought that he could be proud of the man in the video clip, but the video clip suggests otherwise
label: False

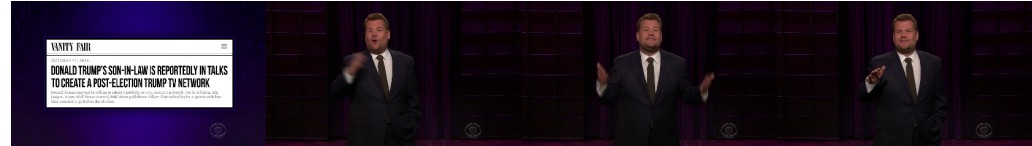

Question: What causes the man speaking to make quotation marks with his hands.
Answer: The man speaking makes quotation to the man on the phone to show that he is not interested in him
label: False

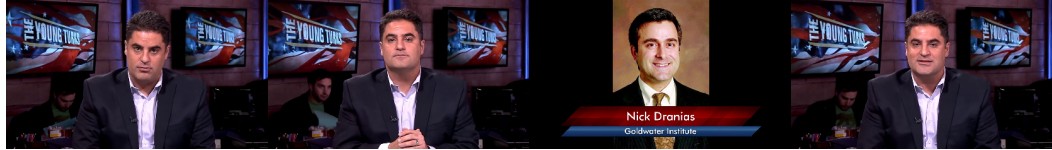

Question: Why does the man make quotation marks with his fingers when he says the word "reporting" at the end of the video?
Answer: The man speaking makes quotation to the man on the phone to show that he is not interested in him
label: False

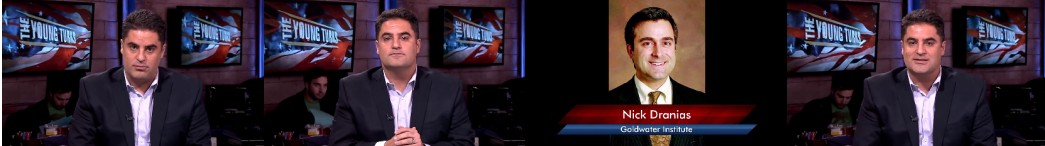

Figure 1: Samples with high answer perceptual score.

Question: What is the tone of the man on the left?
Answer: His tone is very scornful of others.
label: True

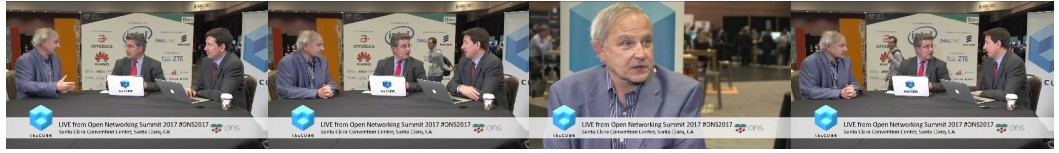

Question: What does the man feel towards the people he mentioned in the speech such as John McCain?
Answer: He hates them all.
label: False

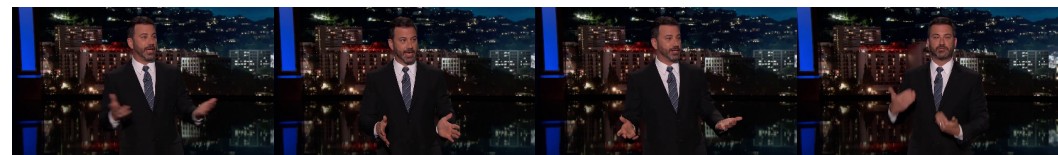

Question: How did some of the people react when the man with glasses told them about the talk show?
Answer: They congratulated him, wishing him luck on it.
label: False

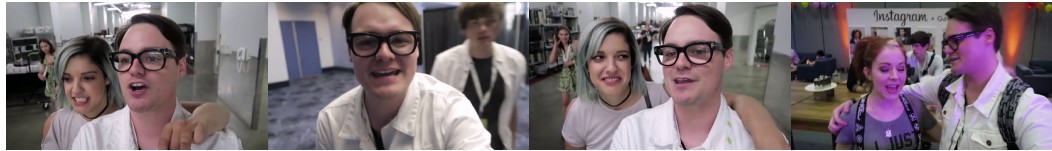

Question: What is the dynamic between the 3 actors?
Answer: The three people are campers and camp master.
label: True

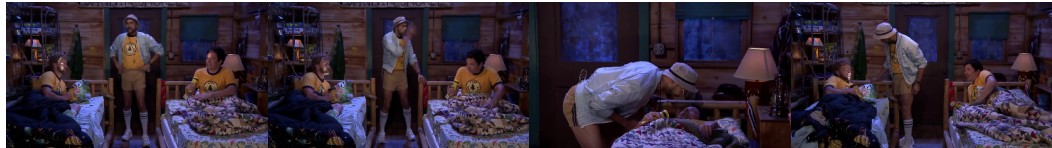

Question: Is the man engaged in what the woman is saying?
Answer: No, he looks bored and unattentive
label: False

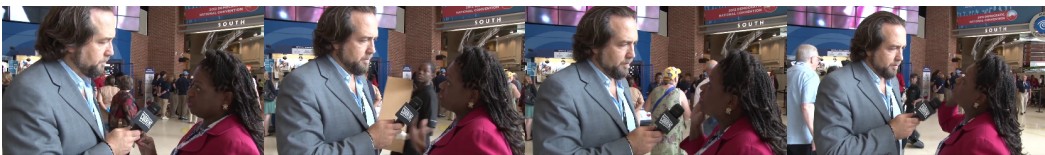

Question: Are the two people sad?
Answer: No, because they are only bored or angry.
label: True

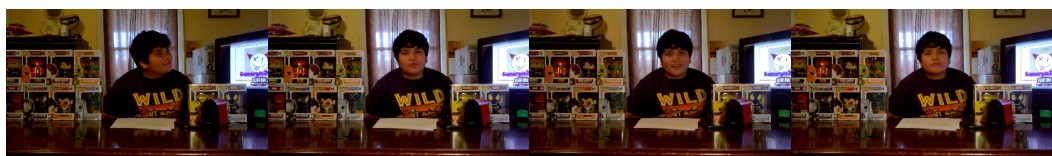

Question: Why is the man nodding while he is speaking?
Answer: He is nervous and fidgety.
label: False

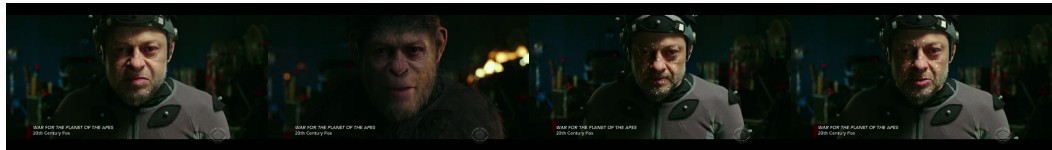

Question: What is the tone of the man with the black suit?
Answer: He is bored but angry
label: False

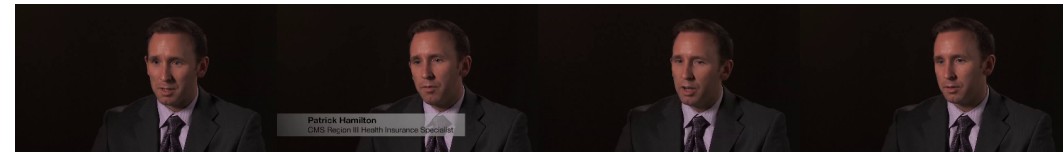

Question: Why did the blonde man lean his head against his fist?
Answer: He was bored and tired
label: True

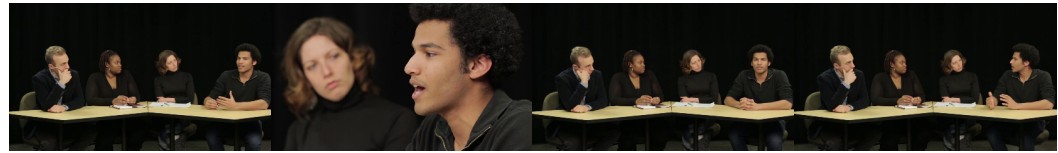

Question: What is conveyed by the posture of the two girls?
Answer: They are hostile and unfriendly.
label: False

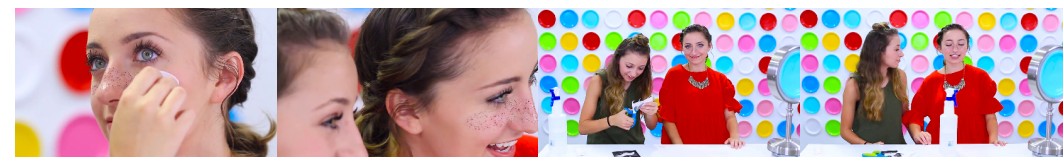

Question: What does the girl think about her service dog?
Answer: She hates her dog for dragging her out of situations she wants to be in.
label: False

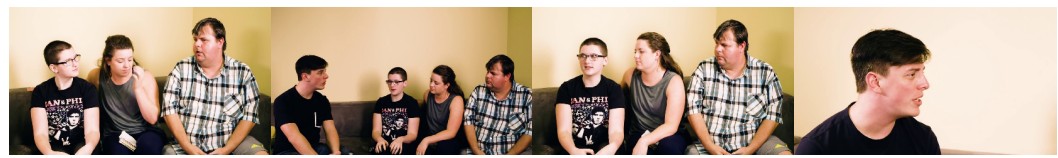

Question: Describe the mood of the man on the left at 0:36
Answer: Sad and angry
label: False

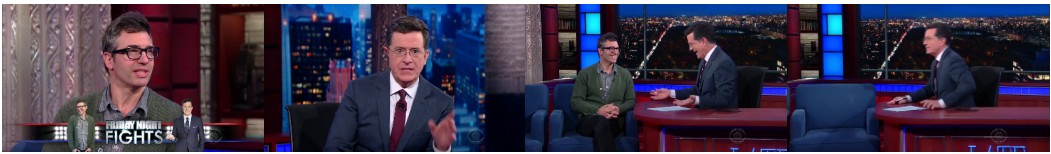

Question: How does the man in the black suit feel about the band?
Answer: He dislikes them; he is actually being sarcastic when he says they are a great band
label: False

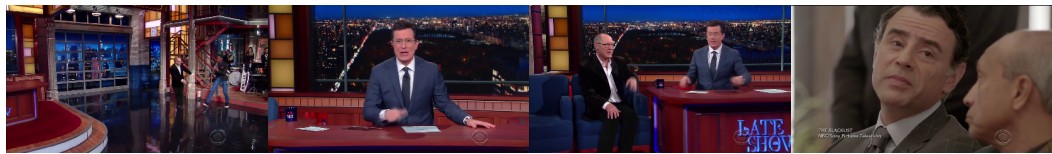

Question: How do the two characters feel at 0:20?
Answer: They are sad and dissappointed
label: False

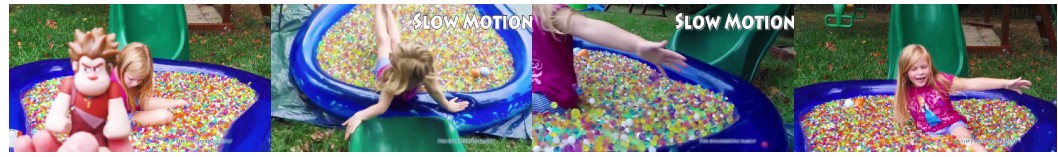

Question: How does the lady on the left feel about commentating?
Answer: She looks bored as a commentator.
label: False

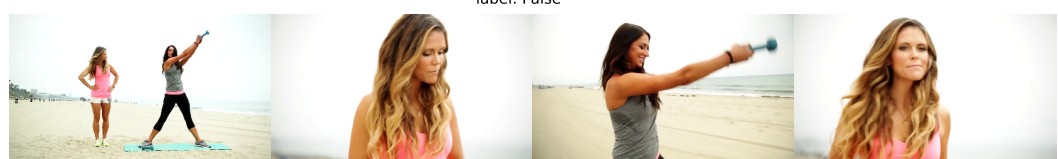

Question: What tone is the woman speaking in?
Answer: She speaks with a rude and opinonated tone.
label: False

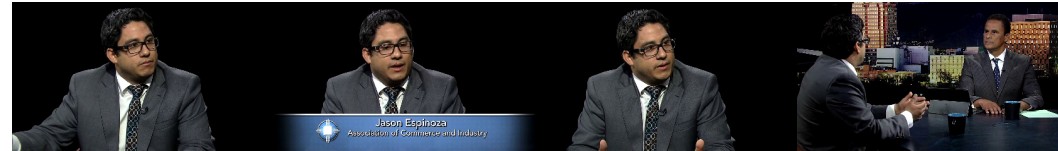

Question: Describe the overall mood of the people in the video
Answer: Full of anger and bitterness
label: False

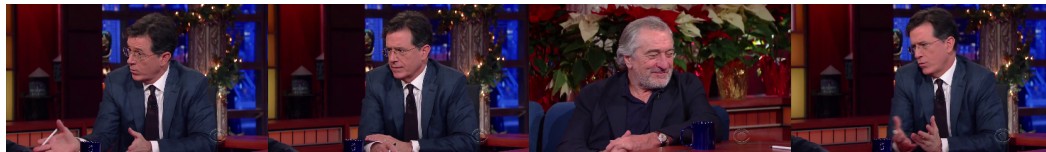

Question: How did the trip make the lady feel?
Answer: She was gloating about her expansive lifestyle.
label: False

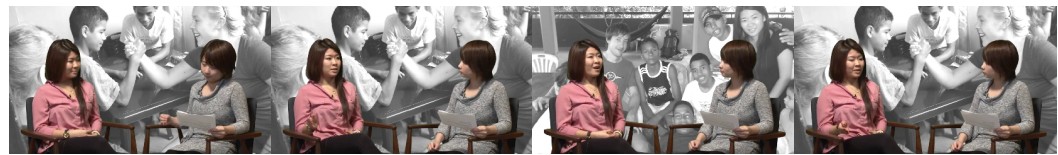

Question: What is the mood of the man at the end of the video?
Answer: Sad and angry
label: False

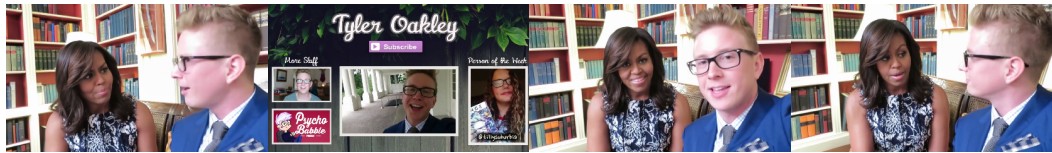

Question: Why does the judge in grey long sleeve cross his arms?
Answer: He appears bored and tired
label: True

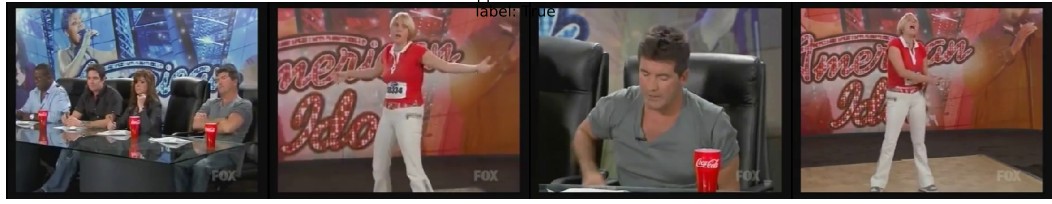

Question: What is the tone of the man at the start of the video?
Answer: Sad and angry
label: False

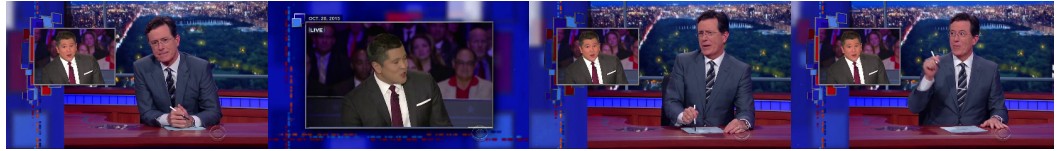

Question: How does the person in the middle feel about the system?
Answer: They are bored and uninformed
label: False

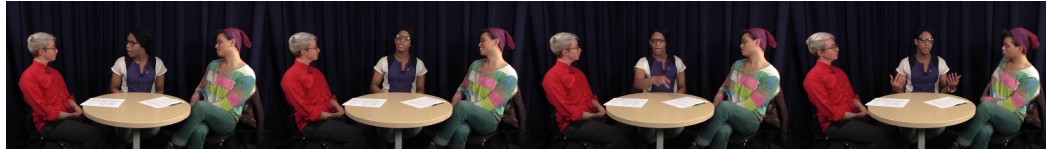

Figure 2: Samples with low answer perceptual score. Note that answer sentiment correlates with the label. Hence a classifier is able to predict the label by detecting sentiment as opposed to analyzing the video (see Sec. 4). For instance, in the first row the answer contains the negative word "hates," and the label is $False$. In contrast, in the second row the answer contains the positive word "energetically," and the label is positive.

# References