# OpenReview forum: "Perceptual Score: What Data Modalities Does Your Model Perceive?"
_NeurIPS.cc/2021/Conference — NeurIPS 2021 Poster_

### Official Review · Reviewer_bqxx · 2021-07-11

**Rating:** 6
**Confidence:** 4

**Summary:**

The paper proposes a score to understand biases in datasets and models by trying to assess the reliance of a model on different modalities. Specifically, this perceptual score is defined as the difference between the model’s accuracy with all modalities and with one modality removed. The proposed method is  used to bring insights into significance of text and visual modalities for different question answering systems.

**Limitations And Societal Impact:**

The paper proposes methods to help understand deep multi-modal models. The authors themselves have outlined two limitations in their conclusions, although they have not been discussed in detail.


**Main Review:**

The paper proposes a method to understand deep neural network models. In particular, multimodal systems are often hard to interpret and the significance of different modalities in solving the task at hand is often not well understood. This is clearly an important problem and in that the paper is clearly well motivated in attempting to do that.
The paper is clearly written and mostly easy to follow.

The current definition of perceptual score and its role in understanding the significance of different modalities as well as biases in models is not very convincing.

1. The perceptual score is showing how the overall accuracy is changing. However, its unclear what this means at sample level. How do we interpret prediction label flip (from correct to incorrect as well as incorrect to correct)  at sample level when “removing” a modality.


2. The modalities are not really removed and I agree that it would be difficult to do for most systems. However, randomly swapping the modality from another data point in the test set is not convincing as well. How do we understand this randomness or stochastic nature of the perceptual score ? Moreover, using the modality input from another data point biases it towards the dataset again. Wouldn’t it be better to just use random “noise” to remove a modality -- say for example some sort of gaussian-noise image input to “remove”. Can this work ? One can also use random image or text from a different dataset.


3. It is unclear how  including Z_D as a normalizing factor helps include the difficulty of a task in the perceptual. Can the authors clarify how  normalization by this factor is really helpful in grounding the perceptual score. If the training data consists of almost even distributed classes then \hat{Acc}_D is basically the 1/#classes which can be large or small depending on the number of classes.


4. The empirical analysis in the paper has focused on using the metric to draw insights into different tasks and systems. However, some empirical analysis on why the metric itself makes sense is necessary ? How do we understand that score of this form is a reliable way to understand datasets and models ?  The paper does not compare or even relate the proposed perceptual score directly to any prior methods which makes it even harder to understand its significance.


5. I am not able to fully understand how the perceptual score aids the bias analysis of CSS. From Table 2, it does look like that CSS is achieving better performance by balancing class priors which are changing from train to test.  But what is the role of perceptual score here ? The numbers in Table 2 and conclusions from it can be drawn without the perceptual score.   Do we need the perceptual score to do this analysis ?



**Time Spent Reviewing:**

5

---

> ### Author Response · Authors · 2021-08-10
> **Response to reviewer bqxx**
>
>
> **Re. it's unclear what this means at sample level. How do we interpret prediction label flip (from correct to incorrect as well as incorrect to correct) at sample level when "removing" a modality.**
>
> In order to "remove" a modality, we permute the modality-related features using data from other samples in the batch. This is repeated multiple times, and the score is computed as an expectation. We measure whether the prediction was correct for every sampled permutation. Note that this doesn't "remove" a modality. Instead, we are permuting one of the modalities using valid instances (instances on the same data manifold) of this data modality obtained from other data points. By permuting the modality, we observe both higher and lower accuracy. What we are interested in is the average effect of this permutation, i.e., how much (typically) lower is the accuracy on average. If a model doesn't rely on a data modality, we would expect hardly a drop in the accuracy (intuitively, the model doesn't perceive/rely on this data modality). Conversely, if a model relies heavily on a data modality, random permutation would result in a significant drop in the predicted accuracy most of the time. This is exactly what the perceptual score intends to capture.
> _________________
>
> **Re. How do we understand this randomness or stochastic nature of the perceptual score?**
>
> Adding means and standard deviations is important when reporting perceptual scores, as we did in our submission. We find the expectation to converge quickly and to be stable. For instance, in Tab. 1, the average standard deviation of the visual perpetual score is 0.36 on VQAv2. Therefore, although the score is stochastic, permutation is stable and converges to similar numbers quickly.
> _________________
> **Re. Using the modality input from another data point biases it towards the dataset again, can we use random image or text from a different dataset?**
>
> Great question. We find it is particularly important to keep the deep net input on the same manifold as the original data that the deep net was trained for.
>
> The use of features outside the expected distribution of the network can cause the network output to be distorted and the performance to be affected due to statistical shifts that harm the information extracted from non-perturbed data modalities. Use of images or text from a different dataset would likely be fine too. However, note that some datasets may have overlapping images.
> _________________
> **Re. Use random "noise" to remove a modality**
>
> Indeed, we also examined random noise as a means to perturb a data modality. However, we found that a careful tuning of the standard deviation of the employed noise is required and sometimes not even possible. Intuitively, if the noise magnitude is too small, results are not affected. If the noise magnitude is too large, the output of the deep net is useless because intermediate representations "overflow".
>
> The fact that permutation of data-modalities is parameter-free while keeping the permuted modality on the original data-manifold was the deciding factor that benefits permutations.
> _________________
> **Re. Why is task normalization helpful?**
>
> Task normalization is particularly useful if a trivial classifier's accuracy is high. Without normalization, the use of permuted data won't modify the accuracy much because the task can be addressed with any data modality. Hence, normalization by the majority vote accuracy amplifies the score difference as we compare the obtained gain (classifier accuracy minus permuted classifier accuracy) to the maximally possible gain (one minus majority vote classifier accuracy).
>
> In the case of evenly distributed classes, a random guess is likely to be incorrect with larger #classes. Hence, the accuracy gain is smaller because the permuted classifier accuracy drop is limited to the one of majority vote.
>
> To make the difference concrete, let's consider the task of VQAv2 in Tab. 1, and let's discuss task normalization. Without normalization, evaluating the other-type questions, we obtain a perceptual score for the visual modality ($P_V$) of LXMERT, which is $P_V$ = 44.05. For Y/N-type questions, $P_V$ = 20.72. This difference suggests LXMERT does not perceive the image for Y/N-type questions.
>
> However, note that even a random guess can achieve 50% accuracy on Y/N questions, i.e., a trivial classifier's accuracy is high. Thus, $\text{Acc}_{\mathcal{M} \setminus V}$ remains high, leading to a presumably low perceptual score of the visual modality. With task normalization, the perceptiveness of other-type and Y/N-type is approximately the same  (41.02 vs. 45.63), which is to be expected.
>
> To summarize, intuitively, task normalization considers the ideal gain (similar in spirit to the NDCG metric, see [2]).  We suggest the ideal gain to be the gap between perfect performance (i.e., 1 in the case of classification) and a trivial prior classifier (i.e., majority vote).
> _________________
> **Re. Why is the score a reliable way to understand datasets and models?**
>
>
> At a high level, our metric follows the intuition of Shapley values. I.e., removing a player (i.e., a modality) and assessing the worth of the coalition (i.e., multimodal classifier) without it. Additionally, the perceptual score is a stable measurement. As we mentioned above, the standard deviation of our metric is low (e.g., in Tab.1, the std-dev of $P_V/Z_{\mathcal{D, f}}$ on average is 0.36). We'll clarify in the revised version.
>
> Despite this, we would appreciate any analysis suggestions.
> _________________
> **Re. Why is the work significant, can we compare prior methods?**
>
> We think our contribution is significant:
> 1. Using the proposed perceptual score, we are able to detect unexplored bias in SocialIQ, and we are able to analyze state-of-the-art methods, such as CSS for VQA-CP. Moreover, the perceptual score is also able to validate the hypotheses of other researchers: specifically, in our response to reviewer NSvs we study an audio-visual benchmark [1]. Hu et al. [1] hypothesize that scene audio can improve the prediction for low-quality images. Using the perceptual score, we can verify this hypothesis by quantifying the degree to which a model perceives the audio data when low-quality and when high-quality images are provided. The perceptual score confirms the hypothesis of Hu et al. [1]: for high-quality images, the audio modality is mostly ignored (the task-normalized audio perceptual score is 9.22); for low-quality images, the audio modality perceptiveness is significant (the task-normalized audio perceptual score is 23.91).
> 2. To date, the accuracy of multimodal models is the primary metric, which we think is no longer sufficient on its own for a proper assessment of a model's behavior. For example, the CSS model achieved high accuracy, but the perceptual score suggests it wasn't necessarily due to better image perception. Please also see our reply below.
>
> Regarding a comparison, we are not aware of any method that specifically addresses the assessment of the perceptiveness of a model towards different data modalities. Although we are unaware of prior work which measures how much a model perceives a modality, we are aware of various works which address the multimodal perceptiveness bias. However, these works focus on altering the dataset (e.g., VQA-CP) or on crafting better models (e.g., CSS). None of these works study a new metric.
> _________________
> **Re. how the perceptual score aids the bias analysis of CSS. Do we need the perceptual score to do this analysis?**
>
> The perceptual score allows us to conduct our study in Tab. 2.
> 1. Despite the high accuracy on Y/N-type questions, the perceptual score reveals that the CSS model does not perceive the image. Thus, the model can derive the answer based solely on the textual modality. For this reason, we focus on the textual input in Tab. 2.  Next, instead of looking at all the Y/N-type questions, we examine the samples that received a low image-perceptual score, which we report in Tab. 2.
>  2. We measured the perceptual score at the individual sample level, which was one of our goals. In Tab. 2, we examined question-types that received low sample perceptual scores. For instance, the token 'has' got an average sample perceptual score of 59.79 and 0.09 for the question and visual modalities, respectively, while the average sample perceptual score across the entire dataset is 39.27 and 0.59. We noticed that the CSS classifier predicted 'yes' for all the 'has'-type questions in the test set, while LMH only answered yes for 47 of the 'has'-type questions (see the first line in Tab. 2). Our attention was then drawn to the prior shift in the train set, which we hypothesized to be the cause for the shift in predictions. Consequently, as we mention in L269, we trained another model by modifying the priors without altering the image as CSS suggests and gets a similar score (57.54 vs. 57.89).
>
> Thus, we think CSS's impressive performance on VQA-CP is at least partly due to the data distribution being shifted closer to the test set during training.
>
> Please note that our study of the SocialIQ bias used samples with low perceptual scores as well. In this case, we found that the label of samples with low perceptual scores correlated with the sentiment of the answers.
>
> In summary, we think the perceptual score is an insightful metric that can be used to reveal different types of biases in either datasets or models.
>
> _________________
> **Reference**
>
> [1] Hu et al. Does Ambient Sound Help? - Audiovisual Crowd Counting. Sight and Sound workshop, CVPR’20.
>
> [2] Cumulated Gain-Based Evaluation of IR Techniques; Jarvelin and Kekalaien; ACM Trans. Inf. Syst. 2002.

---

> > ### Comment · Reviewer_bqxx · 2021-08-18
> > **Thanks**
> >
> > Authors have clarified some of my comments/concerns and I have raised the score to reflect that. One point in particular does require more thorough analysis though. Justification of the metric itself is still desirable. This can be done by comparing/contrasting against some known metrics or by providing some theoretical/empirical justification on why the metric itself makes sense. It has primarily been applied to different datasets.

---

> > > ### Author Response · Authors · 2021-08-21
> > > **Thank you**
> > >
> > > Thank you very much for considering our rebuttal and for taking the time to read it.
> > >
> > > In the paper, we justified the metric by discovering unknown insights on real-world datasets (VQA, VQA-CP, SocialIQ, and VisDial). In the rebuttal (in our response to reviewer NSvs) we further showed that perceptual scores can also be applied to audio data.
> > >
> > > We aren’t aware of existing metrics that provide similar insights for multi-modal classifiers. A comparison to known metrics is hence challenging.
> > >
> > > However, as a sanity check, we can provide even more empirical justification, by studying the perceptual score with synthetic data consisting of three modalities ($a, b, c$). This synthetic data permits control of every aspect of the data. We are happy to include the results below in a revised version.
> > >
> > >
> > > Specifically, we show the perceptual score can identify two properties: (1) utilization of the modality; and (2) informativeness of the modality. In both cases, a low perceptual score is expected if the model does not use a modality or the modality does not provide sufficient information to explain the label. Additionally, if two modalities explain the label equally, their perceptual score should be similar.
> > >
> > > To study this, we generate synthetic data that involves non-linear interactions:
> > > 1. Sample $A \in \mathbb{R}^{d1}$, $B \in \mathbb{R}^{d2}$, $C \in \mathbb{R}^{d3}$ from $U(-\tau,\tau)$
> > > 2. Sample $ \hat{a}, \hat{b} \in \mathbb{R} \sim N(0,1)$ s.t. $|\hat{a} \cdot \hat{b} + \hat{c}| > \delta$
> > > 3. Sample $\hat{c} \in \mathbb{R} \sim N(0, \sigma_c) $, where $\sigma_c \in \mathbb{R}$
> > > 4. If $(\hat{a}\cdot \hat{b}) + \hat{c} > 0$  then label $y=1$ else $y=0$
> > > 5. Return the data point $(\hat{a}A, \hat{b}B, \hat{c}C, y)$
> > >
> > > In order to compute a perceptual score, we use 20 permutations. We calculate the score 10 times (i.e., each time using 20 different permutations) and provide its standard deviation and mean. We create ten different datasets by varying the variance of modality $c$, i.e., $\sigma_c$. We consider a logistic regression (i.e., a linear model) and a two-layer neural network with a non-linear ReLU activation. Note, the logistic regression cannot model multiplicative non-linear interactions. Each dataset includes 2k samples, 1k for training, and 1k for testing. We use the following hyperparameters: $d1=2000, d2=1000, d3=100, \delta=0.25, \tau=1$.
> > >
> > > **Utilization:** In the table below, we show that in a linear model, $a$ and $b$ have low perceptual scores $P_a$ and $P_b$ ($P_a=0.39$ and $P_b=0.56$ on average). As modeling of $a$ and $b$ requires non-linearity, this is expected. In contrast, the perceptual score $P_a$ and $P_b$ of $a$ and $b$ is high in a non-linear neural network ($P_a=35.72$ and $P_b=36.8$ on average).
> > >
> > > **Informativeness:** To examine that the perceptual score reflects the informativeness of a modality, we control the variance of modality $c$ by increasing it from 0 to 1. As expected, as the variance of $c$ increases, the modality becomes more informative (the label depends directly on $c$), and the perceptual score of $c$ increases for both the neural network and the logistic regression. E.g., the perceptual score is zero when the variance of $c$ is zero, and $P_c=32.25$ when $c$'s variance is 1.
> > >
> > > Note that in all cases the symmetry property holds, e.g., $a$ and $b$ contribute equally, and the perceptual score is similar (within the margin of error).
> > >
> > > Regarding normalization:
> > >
> > > 1. Trends obviously don’t change.
> > > 2. In the neural network case, the model-normalization is half of the task-normalization. This is expected when the majority-vote accuracy is centered, as the task normalization considers the ideal gain (i.e., $1 -\text{Acc}_{\cal{D}} = 50$). Indeed, this facilitates the comparison of tasks since different tasks have a different ideal gain. Accordingly, in our case, a perceptual score of 50% is equal to the ideal gain and is considered perfect based on task-normalization. Model-normalization, on the other hand, is more appropriate for weak models. However, in this example, the model achieves near-perfect accuracy. Consequently, the model-normalization doesn't affect the perceptual score.
> > >
> > > Logistic regression table:
> > >
> > > |Var(c)|$\text{Acc}_{\mathcal{M}}$|$P_a$|$P_a/Z_\mathcal{D}$|$P_a/Z_{\mathcal{D},f}$|$P_b$|$P_b/Z_\mathcal{D}$|$P_b/Z_{\mathcal{D},f}$|$P_c$|$P_c/Z_\mathcal{D}$|$P_c/Z_{\mathcal{D},f}$|$\widehat{\text{Acc}_{\mathcal{D}}}$|
> > > |:-:|:-:|:-:|:-:|:-:|:-:|:-:|:-:|:-:|:-:|:-:|:-:|
> > > |0|48|-0.9$\pm$0.18|-1.86|-1.87|-2.04$\pm$0.3|-4.22|-4.24|0$\pm$0|0|0|48.3|
> > > |0.1|50.1|-1.39$\pm$0.31|-2.76|-2.78|1.14$\pm$0.29|2.26|2.28|0.65$\pm$0.03|1.29|1.3|50.4|
> > > |0.2|55.8|-0.08$\pm$0.23|-0.16|-0.15|0.59$\pm$0.26|1.17|1.06|5.86$\pm$0.18|11.58|10.5|50.6|
> > > |0.3|63.2|3.65$\pm$0.1|7.39|5.78|1.59$\pm$0.28|3.22|2.51|13.9$\pm$0.21|28.13|21.99|49.4|
> > > |0.4|66|1.34$\pm$0.14|2.66|2.03|1.29$\pm$0.26|2.57|1.95|16.16$\pm$0.27|32.19|24.49|50.2|
> > > |0.5|72.3|0.61$\pm$0.25|1.29|0.85|1.36$\pm$0.16|2.87|1.88|22.12$\pm$0.36|46.56|30.59|47.5|
> > > |0.6|76.5|0.48$\pm$0.2|0.97|0.63|1.02$\pm$0.23|2.06|1.33|27.57$\pm$0.26|55.81|36.04|49.4|
> > > |0.7|76.6|0.74$\pm$0.12|1.43|0.96|0.93$\pm$0.15|1.81|1.22|26.34$\pm$0.3|51.14|34.38|51.5|
> > > |0.8|79.7|0.47$\pm$0.2|0.92|0.59|0.58$\pm$0.16|1.14|0.73|29.94$\pm$0.23|58.58|37.56|51.1|
> > > |0.9|79.6|-0.8$\pm$0.12|-1.68|-1.01|-0.76$\pm$0.15|-1.59|-0.95|30$\pm$0.33|63.15|37.68|47.5|
> > > |1|81.9|0.27$\pm$0.1|0.51|0.32|0.53$\pm$0.15|1.01|0.64|31.54$\pm$0.35|60.41|38.5|52.2|
> > >
> > > Neural network table:
> > >
> > > |Var(c)|$\text{Acc}_{\mathcal{M}}$|$P_a$|$P_a/Z_\mathcal{D}$|$P_a/Z_{\mathcal{D},f}$|$P_b$|$P_b/Z_\mathcal{D}$|$P_b/Z_{\mathcal{D},f}$|$P_c$|$P_c/Z_\mathcal{D}$|$P_c/Z_{\mathcal{D},f}$|$\widehat{\text{Acc}_{\mathcal{D}}}$|
> > > |:-:|:-:|:-:|:-:|:-:|:-:|:-:|:-:|:-:|:-:|:-:|:-:|
> > > |0|100|49.85$\pm$0.42|98.9|49.85|50.1$\pm$0.3|99.4|50.1|0$\pm$0|0|0|50.4|
> > > |0.1|99.9|49.98$\pm$0.17|104.12|50.03|50.02$\pm$0.38|104.21|50.07|0$\pm$0|0|0|48|
> > > |0.2|98.7|47.3$\pm$0.27|96.14|47.92|47.28$\pm$0.22|96.09|47.9|1.54$\pm$0.05|3.13|1.56|49.2|
> > > |0.3|96.2|41.59$\pm$0.59|80.61|43.24|42.62$\pm$0.4|82.61|44.31|6.24$\pm$0.15|12.09|6.48|51.6|
> > > |0.4|96.1|38.21$\pm$0.36|81.13|39.77|37.95$\pm$0.28|80.58|39.49|11.41$\pm$0.13|24.23|11.88|47.1|
> > > |0.5|95.1|33.32$\pm$0.3|61.48|35.04|33.63$\pm$0.28|62.05|35.36|17.47$\pm$0.21|32.24|18.37|54.2|
> > > |0.6|96.7|31.54$\pm$0.33|60.88|32.61|31.18$\pm$0.23|60.2|32.25|22.14$\pm$0.09|42.74|22.9|51.8|
> > > |0.7|95.1|28.34$\pm$0.37|56.34|29.8|28.63$\pm$0.3|56.93|30.11|22.61$\pm$0.24|44.95|23.77|50.3|
> > > |0.8|94.6|26.5$\pm$0.26|52.37|28.01|26.55$\pm$0.26|52.47|28.07|25.93$\pm$0.29|51.24|27.41|50.6|
> > > |0.9|95.3|23.92$\pm$0.4|50.15|25.1|22.73$\pm$0.27|47.65|23.85|29.08$\pm$0.2|60.97|30.52|47.7|
> > > |1|96.7|22.47$\pm$0.21|44.86|23.24|21.84$\pm$0.3|43.6|22.59|32.58$\pm$0.23|65.04|33.7|50.1|

---

### Official Review · Reviewer_SjCY · 2021-07-14

**Rating:** 6
**Confidence:** 4

**Summary:**

This paper aims to quantify the degree to which a model relies on the different subsets of the input features (i.e., modalities), and introduces the perceptual score. In experimental, they find that state-of-the-art multi-modal models for visual question-answering or visual dialog tend to perceive the visual data less than their predecessors.

**Limitations And Societal Impact:**

Weakness:
1.	This manuscript is more like an experimental discovery paper, and the proposed method is similar to the traditional removal method, i.e., traverse all the modal feature subsets and calculate the perceptual score, removing the last subset. The reviewer believes that the contribution of the manuscript still has room for improvement.
2.	The contribution of different modalities of different instances may be different, e.g., we have modalities A and B, some instances with good performance of modality A  which belongs to the strong modality, whereas some instances with good performance modality B which belongs to the strong modality. Equation 3 directly removes the modal subset of all instances. How to deal with the problem mentioned above.


**Main Review:**

1.	The topic to explore the importance of different modalities is interesting and the reviewer thinks that it will bring more interpretability for multi-modal learning
2.	The proposed method is direct and easy to follow. And the experiments find that multi-modal models tend to ignore some modalities while taking shortcuts.


**Time Spent Reviewing:**

3

---

> ### Author Response · Authors · 2021-08-10
> **Response to reviewer SjCY**
>
>
> **Re. The proposed method is similar to the traditional removal method.**
>
> Rather than defining an importance score for any subset of features, our goal is to develop a measurement tool that assesses the importance of each modality. This hasn't been discussed by prior work in the literature. As we demonstrate in our work, applying this technique to existing models reveals many interesting findings that haven't been addressed before. We would be pleased to discuss the differences between our work and specific traditional removal methods if the reviewer can provide such related work.
> _________________
> **Re. The contribution of the manuscript still has room for improvement.**
>
> We would appreciate concrete suggestions to improve our submission. We will gratefully accept any concrete suggestions regarding improvements we can make.
> _________________
> **Re. The contribution of different modalities of different instances may be different.**
>
> Note that we are not removing modalities as suggested by the reviewer's statement, "Equation 3 directly removes the modal subset of all instances". Instead, we are permuting one of the modalities using valid instances of this data modality obtained from other data points. Indeed by permuting the modality, we observe both higher and lower accuracy. What we are interested in is the average effect of this permutation, i.e., how much (typically) lower is the accuracy on average. If a model doesn't rely on a data modality, we would expect hardly a drop in the accuracy (intuitively, the model doesn't perceive/rely on this data modality). Conversely, if a model relies heavily on a data modality, random permutation would result in a significant drop in the predicted accuracy. This is exactly what the perceptual score intends to capture.
>
> Also note, that we aren't mixing different modalities as suggested by "contribution of different modalities." The perceptual score of a particular modality only randomizes across this specific particular modality. Every modality is considered separately. We think the use of $M_m$ in Eq. (3) clarifies this.
>
> If we didn't correctly understand the reviewer's question, please reach out, and we are happy to reply.

---

> > ### Comment · Reviewer_SjCY · 2021-08-17
> > **Thanks**
> >
> > The author answered my question: "We are permuting one of the modalities using valid instances (instances on the same data manifold) of this data modality obtained from other data points." On the other hand, we need to measure the degree of dependence (i.e., perceptual score) of each data on each modality. This step needs to be repeated many times (multiple sampling operation). I am curious whether this will greatly increase the training time? Overall, the author did answer my doubts and I improved my score.

---

> > > ### Author Response · Authors · 2021-08-17
> > > **Thank you**
> > >
> > > Thanks a lot for your time and for considering our rebuttal. Evaluation of permutations runs in parallel during inference. Training time only changes if researchers want to study the stability of the perceptual score itself using models trained with different seeds.

---

### Official Review · Reviewer_mYsn · 2021-07-15

**Rating:** 7
**Confidence:** 4

**Summary:**

This paper proposes a measure for evaluating multimodal models on how much they take into account the various modalities in making their predictions. The proposed "perceptual score" evaluates a model's performance on each example when one of its modalities is randomly sampled from other items in the dataset (an expectation of performance when paired with all possible versions of the modality being tested). The general idea is that if the performance when paired with a random sample is similar to the performance when paired with the ground truth modality, then the model is likely not taking into account that modality when making its prediction.

The paper evaluates the metric on several vision and language tasks (VQA, VQA-CP, SocialIQ, and VisDial) with several interesting findings (e.g., that one can achieve relatively high performance on SocialIQ without even training a model and only considering one of the modalities).

**Limitations And Societal Impact:**

Although brief, I appreciated the discussion of a "false sense of security" these measures will provide. I have similar concerns that these kinds of metrics will be taken as fact (e.g., that high perceptual scores mean a model/dataset don't have any (inductive) bias).

**Main Review:**

Originality: This is my main concern with the paper and why I am giving a score of 5 in my initial review. Hessel and Lee (EMNLP 2020, https://aclanthology.org/2020.emnlp-main.62.pdf) propose a very similar measure for the same problem studied in this paper, called EMAP. The intuition of the two measures is very similar: measuring the performance on a random sample of modalities. I didn't have time to compare both papers exactly, but I would encourage the authors of this paper to not only cite Hessel and Lee 2020, but include space to compare the measures (since the EMNLP paper has been published for a while now, I would expect the authors of this paper to know about it, and am surprised it was not cited). At a surface level, it does seem the measures are slightly different; for example, they compute a single metric for a dataset while this paper proposes to provide metrics for each modality independently (although I am not sure if this is just \hat{f}_T and \hat{f}_V in Eq 3 of Hessel and Lee 2020). The paper in submission also includes discussion of some normalization used to compare models (and tasks?) directly, and additional analysis on several datasets.

I would expect a final version of this paper to include a discussion of how the method proposed in this paper compares with the measure introduced by Hessel and Lee 2020. Even if the measures are equivalent, this paper does offer additional analysis and variants of the measure that may be interesting standing on their own, although it may require significant reframing of the paper.

Quality: I would have liked a more theoretical intuition of the normalization methods (3.2.2). I was also somewhat confused about when you would use the two different kinds of normalization or how to combine them.

The experiments and analysis are quite thorough, and the finding on SocialIQ is very interesting.

Nit: in lines 160--161, "If the accuracy doesn't change, irrespective of whether modality M_m is available or not, the model f *doesn't* perceive the modality M_m" while this is very *likely* the case, I don't think we can conclude that it doesn't perceive the modality at all; after all the metric is an estimate only (expectation over items in the dataset).

Clarity: The paper is easy to follow and understand for the most part. Being more clear about what "bias" means here could be useful, since this term is ambiguous.

Smaller things:
- Table 1, may be useful to have a reminder of whether a high or low perceptual score is good.
- Table 3: since Acc_D is not dependent on the models maybe don't have different numbers in each model row, rather just have its own combined row.

Significance: A measure like this could be useful for researchers working on multimodal problems. The techniques used in this paper to analyze SocialIQ and VQA-CP provide a good framework for people to interpret and deeply analyze the perceptual scores and their implications on a model and dataset. I worry though that there is significant overlap with EMAP that will cause confusion between the two measures, so would appreciate the authors clearing this up.

**Time Spent Reviewing:**

2

---

> ### Author Response · Authors · 2021-08-10
> **Response to reviewer mYsn**
>
>
> **Re. Comparison to EMAP.**
>
>
> We thank the reviewer for suggesting this relevant work. We think EMAP offers a complementary tool that assesses how much a classifier benefits from interactions across different data modalities. To do this, EMAP approximates a multi-modal classifier with classifiers that each depend on only a single data modality. In contrast, we measure the perceptiveness of each modality separately and for every data point independently, i.e., we aim to identify how much a classifier relies on each data modality. We think both works provide valuable information to study a classifier, which is largely orthogonal.
>
> Understanding how much a classifier relies on individual data modalities (done in our work) as opposed to how much classifier benefits from interactions between different data modalities (done in EMAP) is beneficial for detecting bias, as we show in our paper. The proposed method reveals for every individual sample the modalities which aren't perceived. Analyzing the identified problematic samples then permits to discover reasons for this bias, e.g., for CSS on VQA-CP (where we identify shifts in priors) and for the baseline on SocialIQ (where we identify the correlation between the labels and the answer's sentiment).
>
> Additional differences include:
> 1. Our study proposes a concrete method for calculating a perceptual score that takes the task's difficulty and the model's accuracy into account via different normalizations. In contrast, EMAP only discusses how to compute the closest additive classifier.
> 2. EMAP leaves cases with more than two modalities for future research. It is unclear how to extend EMAP to this case, which requires dealing with interactions of higher-order (see ($\text{proj}_t$, $\text{proj}_v$) in Alg. 1 of EMAP). In contrast, we show results for classifiers with up to three data modalities, and our notations are general.
>
> Also, note that  $\hat{f}_T$ and $\hat{f}_V$ in Eq. (3) of EMAP are not equivalent to the perceptual score. Instead, they represent a partial unimodal model. Our permuted accuracy term, shown in Eq. (3) of our work, captures accuracy and inherently differs. In detail, given modalities $(V, T)$, EMAP uses the permuted term of $V$ as an approximation to the unimodal model of modality $T$.  In our study, we measure the perceptiveness of modality $V$. We do so by evaluating the gain of the original model against the permuted version, which captures both unimodal and cross-modal dependence on modality $V$. We'll add this discussion.
>
>
> _________________
> **Re: Theoretical intuition of the normalization methods.**
>
> We study two different normalizations: 1) task normalization; and 2) model normalization.
>
> 1. Task normalization is particularly useful if a trivial classifier's accuracy is high. Without normalization, the use of permuted data won't modify the accuracy much because the task can be addressed with any data modality. Hence, normalization by the majority vote accuracy amplifies the score difference as we compare the obtained gain (classifier accuracy minus permuted classifier accuracy) to the maximally possible gain (one minus majority vote classifier accuracy).
> 2. Model normalization is particularly useful when the model is relatively weak. Without normalization, the use of permuted data again won't modify the accuracy much because little differences are observed irrespective of whether or not permuted data is used. Hence, normalization by the original model accuracy amplifies the score difference as we compare the permuted classifier accuracy to the original classifier accuracy.
>
> To make the difference concrete, let's first consider the task of VQAv2 in Tab. 1, and let's discuss task normalization. Without normalization, evaluating the other-type questions, we obtain a perceptual score for the visual modality ($P_V$) of LXMERT, which is $P_V$ = 44.05. For Y/N-type questions, $P_V$ = 20.72. This difference suggests LXMERT does not perceive the image for Y/N-type questions.
>
> However, note that even a random guess can achieve 50% accuracy on Y/N questions, i.e., a trivial classifier's accuracy is high. Thus, $\text{Acc}_{\mathcal{M} \setminus V}$ remains high, leading to a presumably low perceptual score of the visual modality. With task normalization, the perceptiveness of other-type and Y/N-type is approximately the same  (41.02 vs. 45.63), which is to be expected.
>
> Intuitively, task normalization considers the ideal gain (similar in spirit to the NDCG metric, see [1]).  We suggest the ideal gain to be the gap between perfect performance (i.e., 1 in the case of classification) and a trivial prior classifier (i.e., majority vote).
>
> Let's now also consider an example for model normalization for the task of VQAv2 in Tab. 1:
> Without normalization, the $P_V$ score of LMH is 26.43 versus 32.51 of LXMERT when considering all questions.
>
> However, note that the original accuracy of LMH is 54.33 while that of LXMERT is 68.97. Also, note that the accuracy drop of LMH relative to its original score (27.90 vs. 54.33) is more significant than that of LXMERT (36.46 vs. 68.97), suggesting that LMH relies more on the image (its drop is slightly larger).
>
> To accommodate for this, the model normalization takes the original model accuracy into account, yielding a normalized $P_V$ score of 48.64 and 47.16 for LMH and LXMERT, respectively, suggesting indeed that LMH relies slightly more on the visual modality.
>
> In conclusion, the perceptual score alone may lead to incorrect conclusions without these normalizations. Therefore, as noted in the paper in L183, when assessing the perceptiveness of the model, we suggest considering both scores.
> _________________
> **Re. The metric is an estimate, change the wording of "doesn't perceive the modality at all."**
>
> Thanks, we agree, and we'll revise our wording.
>
> _________________
> **Reference**
>
> [1] Cumulated Gain-Based Evaluation of IR Techniques; Jarvelin and Kekalaien; ACM Trans. Inf. Syst. 2002.

---

> > ### Comment · Reviewer_mYsn · 2021-08-16
> > **Thanks**
> >
> > Thanks, this clarifies a lot for me. It seems EMAP is still quite relevant to this problem, though distinct; it would be good if the authors could add a small discussion (basically a summary of the comment above) comparing with EMAP. I will raise my score as this addresses my main concern.

---

> > > ### Author Response · Authors · 2021-08-17
> > > **Thank you**
> > >
> > > Thanks a lot for your time and for considering our rebuttal. We will add a summary of this discussion to the main paper and the entire discussion to the supplementary material.

---

### Official Review · Reviewer_NSvs · 2021-07-17

**Rating:** 8
**Confidence:** 4

**Summary:**

This paper pays attention to analyzing the perceptiveness of multi-modal classifiers, proposing a perceptual score, which is very helpful for the community. Using the proposed perceptual score, the authors analyze a wide range of models on different datasets, finding that multi-modal models tend to ignore some modalities while taking shortcuts. In terms of bias, this paper finds that SocialIQ is biased by sentiment. The above interesting findings are helpful for researchers to propose new models and develop new datasets.

**Ethics Review Area:**

["I don’t know"]

**Limitations And Societal Impact:**

1. some models that use more information should be analyzed, such as Oscar [1].

2. this paper only analysis the vision-language datasets, but there are some datasets composed of other modalities (vision, audio and language).

[1]. X. Li et al.. Oscar: Object-Semantics Aligned Pre-training for Vision-Language Tasks. ECCV, 2020.

**Main Review:**

The proposed perceptual score is simple and can be easily computed, which could be a useful metric to measure the perceptiveness of multi-modal classifiers. The perceptual score is novel to me. The analysis is comprehensive, including many existing models on several multi-modal datasets. And the findings are inspiring, which is helpful for the community.

**Time Spent Reviewing:**

2

---

> ### Author Response · Authors · 2021-08-10
> **Response to reviewer NSvs**
>
> **Re. Analyzing Oscar model.**
>
> Thanks for suggesting this work.  In the table below, we show the perceptual scores for Oscar. We find it to be similar to LXMERT, i.e., both score ~47 for the model-normalized visual perceptual score. For the question perceptual score, Oscar is slightly better (61.79 vs. 60.26).   We attribute this similarity to their similar transformer-based architecture. We’ll add these results to the camera-ready version.
>
> | $\text{Acc}_{\mathcal{M}}$ | $\text{Acc}_{\mathcal{M}\setminus \{V\}}$ | $P_V$ | $P_V/Z_{\mathcal{D}}$ | $P_V/Z_{\mathcal{D},f}$ | $P_Q$ | $\text{Acc}_{\mathcal{M}\setminus \{Q\}}$ | $P_Q/Z_{\mathcal{D}}$ | $P_Q/Z_{\mathcal{D},f}$ | $\widehat{\text{Acc}_{\cal D}}$ |
> |:----------------------------:|:-------------------------------------------:|:-------:|:-----------------------:|:-------------------------:|:-------:|:-------------------------------------------:|:-----------------------:|:---------------------------------:|:---------------------------------:|
> | 69.37                      | 36.31                                     | 33.06 | 48.21                 | 47.65                   | 26.93 | 42.44                                     | 61.88                 | 61.79                           | 31.42                           |
>
> _________________
> **Re. Audio modality.**
>
> As suggested, we further investigate a model on the audio-visual crowd counting task [1]. Given an image and an utterance, the task is to count the number of people in a scene. Hu et al. hypothesize that in extreme cases (i.e., low-quality images), the scene audio can improve the prediction [1]. As expected, the perceptual score is able to quantify the degree to which each modality is perceived. We compare the low-quality and high-quality images by employing task normalization. It reveals that in high-quality image settings, the audio modality is mostly ignored (the task-normalized audio perceptual score is 9.22). In contrast, the audio modality perceptiveness in low-quality images settings is significant (the task-normalized audio perceptual score is 23.91). We’ll add these results to the camera-ready version.
>
>
> | Image quality | $\text{1 - MAPE}_{\mathcal{M}}$ | $\text{1 - MAPE}_{\mathcal{M}\setminus \{A\}}$ | $P_A$ | $P_A/Z_{\mathcal{D}}$ | $P_A/Z_{\mathcal{D},f}$ | $\text{1 - MAPE}_{\mathcal{M}\setminus \{V\}}$ | $P_V$ | $P_V/Z_{\mathcal{D}}$ | $P_V/Z_{\mathcal{D},f}$ | $\widehat{\text{1 - MAPE}_{\cal D}}$ |
> |----------------------|:-----------------------------:|:--------------------------------------------:|:-------:|:-----------------------:|:-------------------------:|:--------------------------------------------:|:-------:|:-----------------------:|:-------------------------:|:----------------------------------:|
> | High  | 77.71                       | 73.88                                      | 3.83  | 9.22                  | 4.93                    | 43.16                                      | 34.55 | 83.21                 | 44.46                   | 58.48                            |
> | Low   | 72.51                       | 66.58                                      | 9.93  | 23.91                 | 13.69                   | 45.07                                      | 27.44 | 66.08                 | 37.84                   | 58.48                            |
>
> _________________
> **Reference**
>
> [1] Hu et al. Does Ambient Sound Help? - Audiovisual Crowd Counting. Sight and Sound workshop, CVPR'20.

---

> > ### Comment · Reviewer_NSvs · 2021-08-17
> > **Thanks**
> >
> > Thanks for the response. The experiments on the audio modality seem reasonable. Hopefully, the author can release the toolbox, so that researchers in the community can use it to analyze multimodal tasks and models.

---

> > > ### Author Response · Authors · 2021-08-17
> > > **Thank you**
> > >
> > > Thanks a lot for your time and for reading our rebuttal. We will release a toolbox which the community can use to analyze multimodal tasks and models.

---

### Decision · Program_Chairs · 2021-09-27

**Decision:**

Accept (Poster)

**Comment:**

Understanding how sensitive multi-modal models are to their various inputs is an important topic and this submission proposes a straight-forward to estimate these perceptual sensitivities. The proposed method is demonstrated in a couple of domains and the author response extended to an additional setting. Reviewers unanimously recommend acceptance and the AC concurs. Authors are strongly encouraged to include the additional audio setting in the camera ready and to clarify the comparison to EMAP.